# The genetic interaction map of the human solute carrier superfamily

Gernot Wolf [1,5], Philipp Leippe [1,5], Svenja Onstein [1], Ulrich Goldmann [1], Fabian Frommelt [1], Shao Thing Teoh [1], Enrico Girardi [1,4], Tabea Wiedmer [1] & Giulio Superti-Furga [1,2,3]✉

## Abstract

Solute carriers (SLCs), the largest superfamily of transporter proteins in humans with about 450 members, control the movement of molecules across membranes. A typical human cell expresses over 200 different SLCs, yet their collective influence on cell phenotypes is not well understood due to overlapping substrate specificities and expression patterns. To address this, we performed systematic pairwise gene double knockouts using CRISPR-Cas12a and -Cas9 in human colon carcinoma cells. A total of 1,088,605 guide combinations were used to interrogate 35,421 SLC-SLC and SLC-enzyme double knockout combinations across multiple growth conditions, uncovering 1236 genetic interactions with a growth phenotype. Further exploration of an interaction between the mitochondrial citrate/malate exchanger SLC25A1 and the zinc transporter SLC39A1 revealed an unexpected role for SLC39A1 in metabolic reprogramming and anti-apoptotic signaling. This full-scale genetic interaction map of human SLC transporters is the backbone for understanding the intricate functional network of SLCs in cellular systems and generates hypotheses for pharmacological target exploitation in cancer and other diseases. The results are available at https://re-solute.eu/resources/dashboards/genomics/.

**Keywords** CRISPR Screens; Functional Genomics; Genetic Interactions; Metabolism; Solute Carrier Transporters
**Subject Category** Genetics, Gene Therapy & Genetic Disease

See also: T Wiedmer et al; U Goldmann et al & F Frommelt et al

## Introduction

The relationship between genotype and phenotype is a fundamental question in biology (Orgogozo et al, 2015). Despite the sequencing of the human genome and subsequent genome-wide association studies linking thousands of genetic loci to various traits and diseases, predicting phenotypes from genotypes remains difficult (Dowell et al, 2010). This "genotype-to-phenotype problem" predicament highlights the complex nature of the genetic architecture underlying most biological traits. Increasingly, it has become clear that the effects of genetic variants often depend on both genetic background and environmental context, manifesting as genetic interactions (Domingo et al, 2019). As such, the interplay of genes is fundamental in shaping the genotype-phenotype relationship.

A genetic interaction occurs when the combined effect of mutations in two or more genes results in a distinct phenotype that deviates from the expected additive effects of the individual mutations (Costanzo et al, 2019). These can be classified as either negative, where the combined mutations result in a more severe phenotype than expected (i.e., synthetic lethality), or positive, where the combined mutations lead to a less severe phenotype than expected (i.e., synthetic viability). Genetic interactions are believed to play a fundamental role in many biological processes, from evolution and speciation to the development of complex diseases (Phillips, 2008).

Model organisms have been instrumental in developing methods for large-scale genetic interaction analysis. The budding yeast *Saccharomyces cerevisiae*, in particular, has served as a powerful platform for systematic interrogations of genetic interactions, typically using growth rate as the phenotypic readout (Boone et al, 2007). The average yeast gene participates in approximately 100 negative and 65 positive digenic interactions (Costanzo et al, 2016). Overall, about 3% of all possible gene pairs exhibit some form of interaction, a finding that underscores the necessity of widening the focus of functional genomics away from a single-gene centric view and towards a systems perspective of integrated genetic circuits (Costanzo et al, 2016).

This is important because interactions are non-randomly distributed within the genome and give rise to highly organized networks. Genes involved in related biological processes tend to interact and show similar interaction profiles more frequently than would be expected by chance, forming functional modules that can be associated to specific cellular pathways or compartments (Tong et al, 2004). This modular organization of genetic interaction networks has important implications for predicting gene function, as they allow deconvolution of binary networks to infer modules which enable the inference of roles for uncharacterized genes (Costanzo et al, 2019).

[1]CeMM Research Center for Molecular Medicine of the Austrian Academy of Sciences, 1090 Vienna, Austria. [2]Center for Physiology and Pharmacology, Medical University of Vienna, 1090 Vienna, Austria. [3]Fondazione Ri.MED, Palermo, Italy. [4]Present address: Solgate GmbH, IST Park Building, 3400 Klosterneuburg, Austria. [5]These authors contributed equally: Gernot Wolf, Philipp Leippe. ✉E-mail: gsuperti@cemm.at

Environmental conditions impact both single gene function as well as genetic interactions, resulting in gene-by-environment (GxE) and gene-by-gene-by-environment (GxGxE) interactions. Studies in yeast have shown that the global genetic interaction (GI) network is largely robust to environmental perturbations, however, changes in growth conditions can rewire subsets of interactions and reveal novel functional connections between genes (Costanzo et al, 2021). This environmental dependency of interactions highlights the dynamic nature of genetic networks and their ability to adapt to changing environmental conditions (Kuzmin et al, 2018).

The advent of haploid cell-based loss-of-function and CRISPR-based genetic engineering, coupled with the success of experiments in yeast and other model organisms, could have heralded a plethora of genetic interactions screens in human cells to elucidate gene function broadly (Nijman, 2011). However, this has not exactly been the case. While it was possible to identify human genes conferring cellular fitness under specific tissue culture conditions, describing the so-called essentialome, we have not seen large-scale genetic interaction studies in human cells (Wang et al, 2015; Blomen et al, 2015). Obviously, the primary barrier to such studies is the sheer scale required; the number of possible combinations increases quadratically with each additional gene, leading to a combinatorial explosion. This makes human genome-wide combinatorial screens challenging due to technical limitations in cloning, cell culture scalability, and sequencing depth.

Accordingly, most synthetic lethality studies in human cells have had a comparatively narrow focus either on genes interacting with particular oncogenic genetic conformations or with drugs, pathways or individual protein complexes (reviewed in Huang et al, 2020; Kaelin, 2009; Beijersbergen et al, 2017). More comprehensive studies were guided by protein-protein interactions or have used correlation of effects of single genotype-to-phenotype relationships (Kim et al, 2019; Liu et al, 2011). More recent screens have focused on paralog pairs or splice variants (Thompson et al, 2021; Gonatopoulos-Pournatzis et al, 2020). Nonetheless, certain areas of cell biology and biochemistry may be sufficiently small and functionally coherent, to be both attractive objects of study and technically amenable to systematic genetic interaction approaches, including interactions with the environment.

The ensemble of human membrane transporters may represent such a case. As "gatekeepers" between cells and the external environment, transporters control and regulate the movement of molecules across membranes, directly influencing cellular responses to environmental changes. Notably, some transport processes are directly linked to epigenetic states and transcriptional regulation (Li et al, 2021). This makes transporters ideal candidates for investigating the interplay between genetic variation, environmental factors, and phenotypic outcomes (Chidley et al, 2024; Rebsamen et al, 2022).

Among transporters, the solute carrier (SLC) superfamily, comprising 456 genes in humans, stands out as the largest group of transporter genes (Ferrada and Superti-Furga, 2022; Perland and Fredriksson, 2017). This diverse superfamily of transmembrane proteins is responsible for the transport of a wide range of substrates, including ions, metabolites, and drugs (Hediger et al, 2013). Despite their crucial role in virtually all aspects of cell biology, at least a third of the SLC superfamily lacks any functional annotation (Meixner et al, 2020; César-Razquin et al, 2015).

A typical human cell expresses more than 200 different SLCs (Jin et al, 2023; O'Hagan et al, 2018), often with overlapping functions and expression patterns. This redundancy is a necessary feature of cellular transport systems, given the plethora of structurally diverse molecules required to be moved across membranes to sustain cellular needs. For instance, human serum alone contains over 4,600 metabolites (Psychogios et al, 2011), which suggests the necessity of a broad specificity transport system to handle this diverse molecular repertoire with a relatively limited set of genes. This functional overlap among SLCs presents a challenge for functional genomics. Traditional single-gene knock-out studies often fail to reveal the full spectrum of a transporter's function due to compensation by other superfamily members and general metabolic plasticity (Girardi et al, 2020a). Therefore, superfamily-wide, combinatorial strategies may be required to unravel the individual and unique roles of specific SLCs together with their collective functions at systems-level.

Given the complex interplay between genetic interactions, environmental influences, and the unique characteristics of transporters, a systematic approach to investigate the functional relationships among SLCs is warranted. To accomplish this, we here leveraged recent advances in CRISPR technology to perform multiple large-scale genetic interaction screens focusing on SLC transporters (DeWeirdt, 2020; Li et al, 2022). Using both Cas12a and Cas9 systems, we performed five distinct SLCxSLC and SLCxEnzyme combinatorial double knockout screens in the human HCT 116 colorectal carcinoma cell line, the latter under four growth conditions. In total, we measured the growth phenotype over time of 35,421 different double knockouts along with their corresponding single knockout effects. This allowed for data integration from different CRISPR systems and growth conditions, minimizing system-specific biases and providing cross-validation of findings. This study, part of the RESOLUTE consortium's comprehensive efforts to characterize the SLC superfamily (Fig. 1A), complements parallel investigations in metabolomics, transcriptomics (Wiedmer et al, 2025), protein interactions (Frommelt et al, 2025), and multi-omics data analysis (Goldmann et al, 2025). This data set is interactively available at https://re-solute.eu/resources/dashboards/genomics/.

## Results

### Cas12a-Metal-SLCxSLC: metal transporter-focused combinatorial KO screen

We chose an enCas12 CRISPR system because it is able to process two gRNAs from a single transcript, making it ideal for a combinatorial double KO study (DeWeirdt, 2020). To assess its effectiveness in double SLC KO editing, we started with a small subset of transporters. We designed a focused library (Cas12a-Metal-SLCxSLC) targeting all possible pairwise combinations of 21 heavy metal-transporting SLCs that are expressed (>1 transcript per million) in the colorectal carcinoma cell line HCT 116 (Goldmann et al, 2025). Metal transporters were chosen for this benchmarking screen due to their manageable number, high level of redundancy, and crucial role in maintaining metal homeostasis (Kambe et al, 2021; Zoroddu et al, 2019).

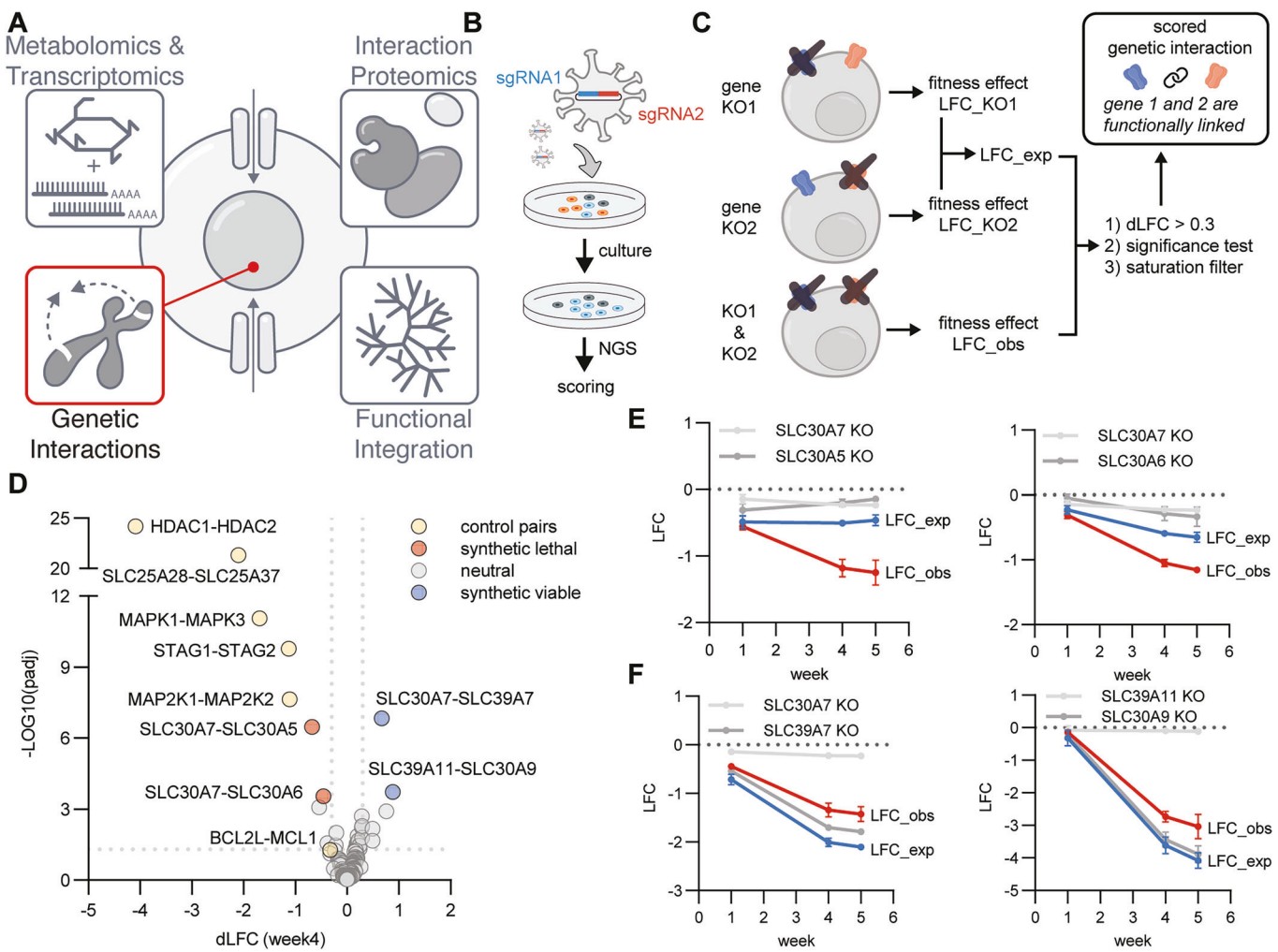

**Figure 1. Workflow for genetic interaction mapping of SLC transporters and results of benchmark screen targeting 21 metal SLC transporters plus controls (Cas12a-Metal-SLCxSLC).**

(A) Overview representation of SLC superfamily-wide data set covered within this RESOLUTE paper collection. (B) Workflow of this genetic interactions study. HCT 116 cells expressing enCas12a or Cas9 were transduced with a lentiviral combinatorial sgRNA library, selected with puromycin, and cultured for up to five weeks. Cells were harvested at multiple time points, followed by library amplification and Illumina sequencing. (C) Abundances of sgRNA combinations at time points were normalized to input, deriving log2 fold changes (LFC) and significance scores (padj) for three replicates. The difference dLFC was derived from LFC_exp, that is the sum of LFCs of single knockouts, and LFC_obs, the experimentally observed LFC. Interactions were scored as genetic interaction if all three replicates tested significant (padj < 0.1), and if |dLFC| > 0.3. Moreover, a saturation filter is applied to minimize false-positive synthetic viable interactions. For example, the SLC39A7-SLC30A9 interaction, showing a positive dLFC but smaller LFC_obs than both single KOs, is filtered out (Fig. EV1A). (D) The metalSLC library screen recovered all control gene pairs (yellow) and identified synthetic lethal (red) and viable (blue) interactions, four weeks post-transduction. Data shown as mean of three independent replicates. Statistical significance determined using Student's paired t-Tests with a two-tailed distribution. P values were adjusted using the two-stage linear step-up procedure of Benjamini, Krieger and Yekutieli. (E) Change of LFC values over time for synthetic lethal interactions (LFC_obs < LFC_exp). Mean ± SD of three replicates. (F) Change of LFC values over time for synthetic viable interactions (LFC_obs > LFC_exp). Mean ± SD of three replicates.

The library design included SLC-targeting sgRNAs (three per SLC) paired with olfactory receptor (OR) sgRNAs to assess single KO effects, and six synthetic lethal gene pairs as positive controls. At gene level, this design resulted in a library size of 210 SLCxSLC pairs. These were cloned into a lentiviral vector for combinatorial Cas12a sgRNA expression (DeWeirdt, 2020), and used in a pooled growth fitness screen in a HCT 116 subclone stably expressing the enCas12a protein (DeWeirdt, 2020) (Fig. 1B). Cells were harvested at multiple timepoints (week 1, 4, and 5), and sgRNA abundance in cell pools was determined by NGS.

To quantify fitness and score genetic interactions, we calculated the difference between the observed log2 fold change (LFC_obs) of

sgRNA pair abundance over lentiviral plasmid library input and the expected value (LFC_exp), defined as the sum of the single KO effects, to obtain the differential log2 fold change (dLFC) for each gene pair as scoring metric (Dataset EV1, Fig. 1C). After applying dLFC (|dLFC| > 0.3) and significance thresholds (padj < 0.1), a fitness saturation filter was additionally applied to minimize false positives, particularly for synthetic viable interactions involving essential genes. For instance, the SLC39A7-SLC30A9 interaction would score as synthetic viable (Fig. EV1A). However, as both genes are essential, it needs to be considered that cells already dying or entering growth arrest due to one KO are unlikely to show a

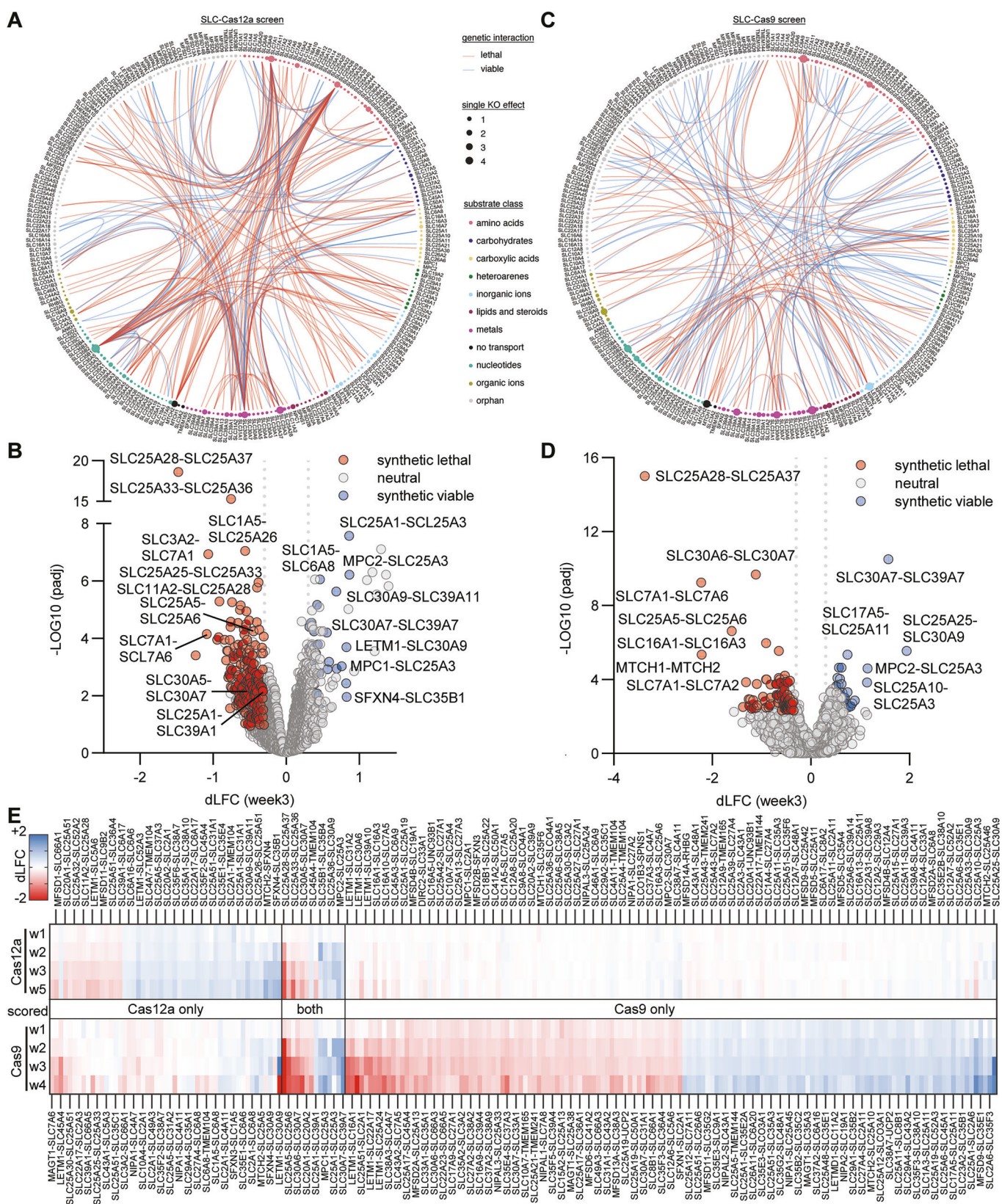

**Figure 2.   Genetic Interactions among SLC transporters in HCT 116 cells.**

(A) An overview of 228 genetic interactions among the 258 SLC transporters expressed in HCT 116 cells, identified through a combinatorial CRIPSR-Cas12a double knockout screen of 33,153 SLC-SLC pairs (SLC-Cas12a). The 258 SLCs are ordered by substrate class. The round circles describe single KO effect sizes. Genetic interactions are shown as connecting bands between SLCs. (B) Volcano plot of the 228 interactions in (A), showing the magnitude dLFC and significance $-\log10(padj)$ of the interactions. Interactions were only scored as significant if all three replicates scored independently as significant (padj < 0.1). Student's paired t-Tests with a two-tailed distribution. *P* values were adjusted using the two-stage linear step-up procedure of Benjamini, Krieger, and Yekutieli. (C) Same as (A) but using an orthogonal CRISPR-Cas9 double KO system (SLC-Cas9), which identified 169 genetic interactions. (D) Same as (B), both for the SLC-Cas9 screen in (C). As the Cas9 screen was only performed in two replicates, they were combined before testing significance once with more stringent cutoff (padj < 0.005). Student's paired t-Tests with a two-tailed distribution. *P* values were adjusted using the two-stage linear step-up procedure of Benjamini, Krieger, and Yekutieli. (E) Overlap between the SLC-Cas12a and the SLC-Cas9 data sets, resulting in 208 scored genetic interactions, after exclusion of interactions involving six frequently scoring SLCs that were essential as single KO in both screens based on a LFC(single KO) < −2. The upper band shows the Cas12a dLFC values across timepoints, and the Cas9 values at the bottom.

much stronger phenotype when the second gene is depleted. In the case of SLC39A7-SLC30A9, the combined KO effect was stronger than each of the single SLC30A9 KO effects, but it did not reach the calculated expected effect of the summed up single KO effects. Such synthetic viable interactions were only scored if the combined KO effect was smaller than either of the single KO effects.

All six synthetic lethal control pairs scored as expected, including the well-documented SLC25A28-SLC25A37 interaction (Shi et al, 2022; Girardi et al, 2020a) (Fig. EV1B). Among the 210 tested SLC gene pairs, four interactions (1.9%) were identified (Fig. 1D). Two pairs were synthetically lethal (SLC30A7-SLC30A5 and SLC30A7-SLC30A6), and two were synthetically viable (SLC30A7-SLC39A7 and SLC39A11-SLC30A9).

These findings highlight the utility of genetic interactions for inferring shared functions among SLC family members, particularly within the context of substrate transport. The synthetic lethality observed between SLC30A7, SLC30A5, and SLC30A6, which are all members of the zinc transporter (ZnT) family, suggests functional redundancy in zinc transport. Disruption of both genes in each pair likely impairs the cellular zinc homeostasis to a lethal extent, reflecting their overlapping roles in exporting zinc out of the cytosol. Conversely, the synthetically viable interactions—SLC30A7-SLC39A7 and SLC39A11-SLC30A9—occur between transporters that move zinc in opposite directions. SLC30A7 and SLC30A9 are ZnT transporters that export zinc out of the cytosol, while SLC39A7 and SLC39A11 are ZIP transporters that import zinc into the cytosol. The viability of these pairs suggests that if a disruption in zinc homeostasis occurs due to the knockout of one transporter, the knockout of a second transporter with an opposing function can restore the balance, thereby correcting the growth defect.

By comparing dLFC values over time for synthetic lethal (Fig. 1E) and synthetic viable interactions (Fig. 1F), we optimized harvest time points for subsequent screens, overserving that single and double KO effects plateaued between weeks 4 and 5 (Fig. EV1C). This led us to adjust harvesting time points for subsequent screens to weeks 1, 2, 3, and 5. Taken together, these results demonstrated the robustness of the workflow and of the scoring approach.

## Cas12a-SLCxSLC and Cas9-SLCxSLC: SLC superfamily-wide combinatorial KO screens

Building on the success of the Cas12a-Metal-SLCxSLC screen, we conducted an SLC superfamily-wide screen (named Cas12a-SLCxSLC) targeting all 33,153 possible SLC-SLC pairs among the 258 expressed in HCT 116 cells (Goldmann et al, 2025). The library included OR controls, totaling 617,796 sgRNA combinations, with

18 sgRNA combinations per SLCxSLC pair. We collected cells at weeks 1, 2, 3, and 5 post-transduction, and derived dLFC scores as described before for the metal-focused screen. Based on week 3 data, we identified 228 (0.69%) SLC-SLC interactions; 43 synthetic viable and 183 synthetic lethal (Fig. 2A, Dataset EV2).

This SLC-wide screen successfully recovered all metal transporter interactions (SLC25A28-SLC25A37, SLC30A5-SLC30A7, SLC30A6-SLC30A7, SLC30A9-SLC39A11, and SLC30A7-SLC39A7). observed in the smaller Cas12a-Metal-SLCxSLC screen (Figs. 2B and EV2F). Correlation analysis of dLFC values between the two screens showed a Pearson correlation coefficient (PCC) of 0.91 and $R^2$ of 0.83 at the time point used for scoring genetic interactions (Fig. EV1D), indicating high reproducibility despite the very large library size. Notably, 159/228 of scored interactions involved six SLCs deemed to be essential or semi-essential (SLC25A26, SLC7A1, MTCH2, SLC30A9, SLC25A3, and SLC35B1) based on large single KO LFC effects (Fig. EV2A). Several known interactions, such as SLC16A1-SLC16A3, MTCH1-MTCH2, and SLC2A1-SLC25A51 (Shi et al, 2022; Girardi et al, 2020a), were not detected in this screen. While this discrepancy could be attributed to cell line-specific differences, it might also indicate a high false-negative rate in our screen. This could potentially stem from lower editing efficiencies of Cas12a sgRNAs or the limited number of sgRNAs targeting each SLC.

To investigate the possible reasons for these differences and cross-validate the SLC genetic interaction map, we conducted an orthogonal screen using CRISPR-Cas9 (Cas9-SLCxSLC). This screen also tested all 33,153 possible gene pairs but employed a modified cloning strategy to reduce library complexity while increasing the number of sgRNAs per gene from three to six. This approach resulted in 437,172 sgRNA combinations, providing an independent, complementary data set to our Cas12a screen.

Based on week 3 dLFC values, 169 (0.51%) high-confidence genetic interactions were identified in the Cas9-SLCxSLC screen (Fig. 2C), comprising 93 synthetic lethal and 76 synthetic viable interactions (Figs. 2D and EV2F, Dataset EV3). Except for BCL2L1-MCL1, which also only scored weakly in the Cas12a-SLCxSLC screen, all control pairs showed strong synthetic lethal interactions in the Cas9-SLCxSLC screen (Dataset EV3) Unlike with the Cas12a system, several previously reported interactions including SLC16A1-SLC16A3, MTCH1-MTCH2 and SLC2A1-SLC25A51 were detected (Girardi et al, 2020b; Shi et al, 2022). Importantly, only a small percentage (12/169) of the identified Cas9 interactions involved the six most essential SLCs (Fig. EV2B), possibly due to differences in double editing efficiencies between Cas12a and Cas9.

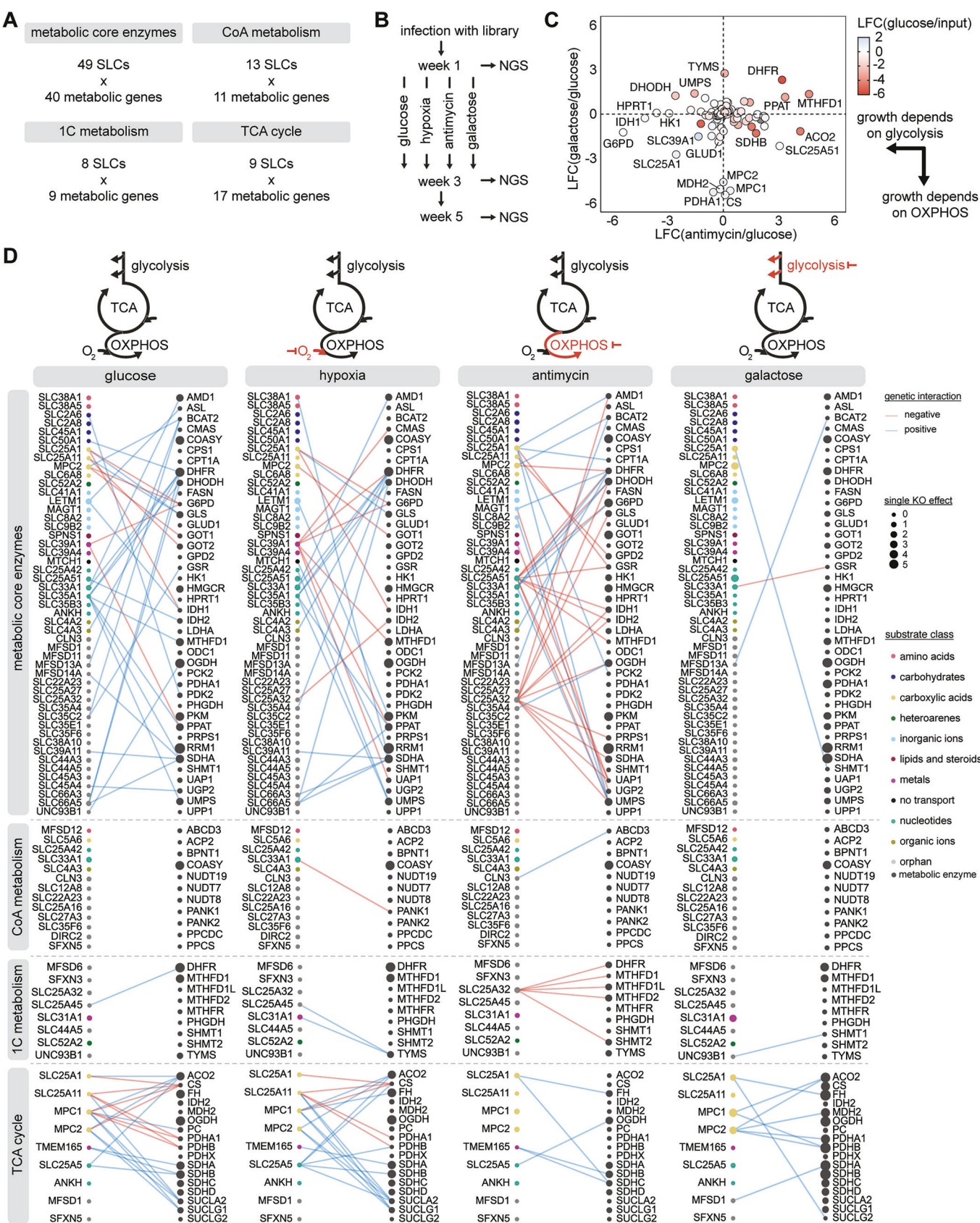

◄

**Figure 3.  Genetic interactions of select SLC transporters with metabolic enzymes (Cas12a-SLCxEnzyme) under different growth conditions.**

(A) Using the Cas12a system, four sublibraries were designed to cover interactions between SLCs and enzymes involved in specific metabolic pathways. (B) To uncover synthetic lethality or viability between SLCs and metabolic enzymes under different conditions, cells were cultured in standard high-glucose media, hypoxia (1% O$_2$), presence of antimycin (50 nM), or media with glucose mostly replaced by galactose (80% → 90% → 80% galactose). Cells were harvested at multiple time points, and dLFC values were derived and genetic interactions scored based on week 3 compared to input plasmid DNA. (C) Overview of single KO LFCs in standard high-glucose medium (glucose/input; blue-red circle color) compared to antimycin (antimycin/glucose, x-axis) or galactose (galactose/glucose, y-axis). (D) Overview of genetic interactions from the four tested sublibraries under the four growth conditions. Circle size encodes the single KO effects which are colored according to substrate class. Blue connections are synthetic viable and red connections are synthetic lethal interactions.

Single KO fitness effects were generally similar in both screens, with a few exceptions such as LETM1 and SLC15A2, but in agreement with regards to the six most essential SLCs (SLC25A26, SLC7A1, MTCH2, SLC30A9, SLC25A3, and SLC35B1; LFC < −2 in both screens). Notably, among these six genes, one (SLC25A26) displayed similar single KO effects in both screens, while four (SLC7A1, SLC25A3, MTCH2, SLC30A9) showed stronger single KO effects in the Cas9 screen (Fig. EV2C), potentially explaining the lower number of interactions involving these genes in the Cas9 screen. The sgRNA orientation in the vector (sgRNA1-gRNA2 vs sgRNA2-sgRNA1) had no effect, with R$^2$ values of 0.96 (Cas12a-SLCxSLC) and 0.99 (Cas9-SLCxSLC) for single KO effects (Fig. EV2D) and 0.82 (Cas12a-SLCxSLC) and 0.83 (Cas9-SLCxSLC) for double KO effects (Fig. EV2E). These double KO correlation values are lower than those previously reported for paralog pairs using Cas12a (R$^2$ = 0.86) and Cas9 (R$^2$ = 0.96) (Li et al, 2022), potentially due to false negatives in our screen.

This suggested that the 163 Cas12a interactions involving essential SLCs may require additional interpretative caution (Fig. EV2G). Excluding synthetic lethal interactions involving the six essential SLCs in both Cas12 and Cas9 screens, a combined total of 208 (0.67%) SLC-SLC interactions were detected, with a PCC of 0.42 between the screens based on the dLFC (week3) values that were used for scoring interactions (Fig. 2E). 14 SLC-SLC interactions were detected in both screens, with all pairs except SLC45A4-TMEM104 sharing either a common substrate or localization (Dataset EV4). Among these, notable examples for common substrate pairs included SLC25A5-SLC25A6 for nucleotides, SLC25A33-SLC25A36 for pyrimidines, and SLC20A1-SLC20A2 for phosphate. Importantly, 7 out of the 14 interactions were between SLCs expressed in the inner mitochondrial membrane, suggesting a potential lower level of functional redundancy among mitochondrial SLCs. Additionally, four synthetic lethal interactions with essential SLCs were observed with both screening systems (SLC25A3-SLC39A1, SLC3A2-SLC7A1, SLC7A1-SLC7A2, and SLC7A1-SLC7A6).

Given our very conservative scoring approach, we consider the interactions robust even if they only scored in one of the two SLC-wide screens. This confidence is supported by the expected behavior of positive control pairs and the high correlation and reproducibility between the two Cas12a screens that were performed and analyzed independently.

## Cas12a-SLCxEnzyme and Cas9-SLCxEnzyme: mapping genetic interactions between SLCs and enzymes in different growth conditions

We next investigated genetic interactions between SLC transporters and metabolic enzymes to explore the functional relationship between transport and cellular metabolism. Previous studies examining individually selected SLCxEnzyme interactions, though limited in scope, have yielded valuable insights (Girardi et al, 2020b; Wei et al, 2022), indicating the potential of a more comprehensive screen. Therefore, we designed a larger SLCxEnzyme library, strategically selecting metabolic enzymes involved in key different metabolic pathways. For SLCs, our selection was informed by our prior SLCxSLC screening results and the 'orphan' status of some SLCs with missing substrate annotation.

Overall, we designed four combinatorial sublibraries covering different aspects of metabolism (Fig. 3A and Dataset EV5). The first sublibrary included 40 metabolic core enzymes paired with 49 SLCs, including 20 orphans, selected based on their potential involvement in these metabolic pathways. The second sublibrary focused on CoA metabolism, pairing 8 SLCs (including two CoA transporters) with 9 metabolic genes involved in CoA homeostasis. The third sublibrary targeted one-carbon (1 C) metabolism, combining 13 SLCs (including three transporters of 1 C metabolism substrates) with 11 metabolic enzymes. The fourth sublibrary combined 9 SLCs (including four transporters of TCA substrates) with 17 TCA cycle enzymes. In total, these four sublibraries tested interactions between 2286 SLC-enzyme pairs, supplemented with a handpicked panel of 54 SLC-SLC pairs, yielding a final library size of 2340 gene pairs. We used a library design combining six sgRNAs per gene in one orientation only to provide six sgRNA pairs per gene pair and reduce library size.

To uncover synthetic lethality or viability with metabolic enzymes, that might only be observable under growth conditions other than standard high glucose medium, we performed the Cas12-SLCxEnzymes screen with varying growth conditions (Fig. 3B). On top of the previously used standard conditions, cells were grown under hypoxia in a 1% oxygen incubator, reducing the oxygen available for oxidative phosphorylation (OXPHOS) and thereby decreasing its rate (hypoxia). The second additional condition (antimycin) included Antimycin A (50 nM), that blocks complex III of the respiratory chain (RC), thus inhibiting OXPHOS and reducing flux through the TCA cycle, making cells more reliant on glycolysis. In the third additional condition (galactose), glucose was replaced with galactose. Cells utilize galactose at lower rates than glucose in glycolysis, making them more reliant on mitochondrial activity to fulfill their metabolic needs. We cloned and used both Cas12a and Cas9 versions of the same library, again for cross-validation and minimizing system specific biases.

After library transduction and puromycin selection, the HCT116-Cas12a clone grew faster than the HCT116-Cas9 clone, with a doubling time of 1.5 days compared to 2.1 days, respectively (Fig. EV3A), strongly suggestive of clonal differences. For the Cas12a cells, hypoxia only had a small effect on growth, but antimycin and galactose presence caused strong proliferation

defects, increasing average doubling times to 2.5 and 2.0 days, respectively. Despite this, the Cas12a-SLCxEnzyme screen was successfully completed under all conditions. In contrast, the transduced HCT116-Cas9 clone showed higher sensitivity to antimycin and galactose, resulting in severe growth inhibition that precluded cell harvesting after the first week. Consequently, the Cas9-SLCxEnzyme screen could only be concluded for the glucose and hypoxia conditions.

We analyzed the KO effects across the four growth conditions in the Cas12-SLCxEnzyme screen. Hypoxia induced minor fitness differences for 11 genes compared to glucose (Fig. EV3B). In contrast, antimycin A and galactose treatments resulted in more pronounced differences for 46 and 36 genes, respectively. These differences revealed clusters of genes connected by cellular localization and metabolic pathways (Fig. 3C). For instance, KOs of transporters and enzymes involved in fueling the TCA cycle with pyruvate (e.g., MPC1/2, MDH2, PDHA1, CS) led to severe growth defects in galactose but slight growth advantages in glucose, while showing no defect under antimycin A treatment.

Another cluster of genes, non-essential in glucose, became highly essential for proliferation in antimycin media. These included genes involved in glycolysis and the pentose phosphate pathway (HK1 and G6PD), and IDH1. The latter's importance likely stems from its role in rescuing cytoplasmic $NADP^+$ regeneration when OXPHOS is inhibited. HPRT1, involved in the purine salvage pathway, likely becomes essential due to impairment of de novo purine synthesis. Other genes with strong environmental lethality or viability included several other players in nucleotide synthesis (DHODH, UMPS, TYMS, PPAT), plus in folate metabolism (DHFR, MTHFD1) and mitochondrial transport (SLC25A1 and SLC39A1). Of the total of 134 observed single KO effects, 41 (31%) changed with |LFC| > 1 when comparing galactose or antimycin with the glucose condition (Fig. 3C).

We then analyzed double KO effects at week 3, finding 56 genetic interactions in standard glucose, 51 in hypoxia, 49 in antimycin, and 13 in galactose condition (Fig. 3D). In glucose media, 56 interactions out of the 2286 possible SLC-enzyme pairs (2.4%) were identified. Interactions between SLCs and TCA cycle enzymes were overrepresented, with 21/153 pairs (12%). The four mitochondrial carriers shuttling TCA cycle metabolites (SLC25A1, SLC25A11, MPC1, and MPC2), exhibited 17 synthetic lethal or viable interactions out of the 68 possible (20%). This is indicative of mitochondrial metabolisms' ability to compensate for loss of single genes by compensatory rerouting of metabolic fluxes, but compensations start failing once two genes are ablated. For instance, when pyruvate anaplerosis is disrupted by ablation of PDHA1/PDHB1, SLC25A11 might compensate by importing cytosolic alpha-ketoglutarate (aKG) in exchange for mitochondrial malate. Once SLC25A11 is ablated as well, this compensatory mechanism fails, explaining the observed strong synthetic lethality.

Consistent with the small impact of hypoxia on single KO effects (Fig. EV3B), the number and pattern of genetic interactions was generally very similar to the glucose condition. A few notable differences included synthetic lethality between the zinc transporter SLC39A1 and the methylenetetrahydrofolate dehydrogenase MTHFD1, and the UTP consuming enzymes CMAS and UAP1, suggesting a connection between SLC39A1 and one-carbon (1C) metabolism. Moreover, while an interaction between the mitochondrial orphan transporter SLC25A45 and DHFR was found in

glucose, SLC25A45 interacted with thymidylate synthetase TYMS in hypoxia. DHFR produces tetrahydrofolate (THF), whereas TYMS consumes it. This change in interaction partners may indicate a reversal in folate cofactor flux, suggesting a role for SLC25A45 in one-carbon metabolism (Carreras and Santi, 1995; Czekster et al, 2011).

Antimycin A treatment caused more drastic changes at single KO level (Fig. 3C) compared to glucose. Consistent with Complex III blockade, TCA cycle enzymes were less essential, and exhibited much fewer genetic interactions (5). The single KO effects of the mitochondrial 1C enzymes MTHFD1L, MTHFD2, and SHMT2 were more lethal, whereas the single KO effects of cytosolic MTHFD1 and DHFR were buffered (Fig. 3C,D). A total of 15 synthetic lethality interactions were observed between those enzymes and the mitochondrial carrier SLC25A32, which had no genetic interactions in other conditions (Fig. 3D). SLC25A32 was reported to import THF into mitochondria (Shi et al, 2022), which potentially compensates for RC inhibition, and may also import flavin adenine dinucleotide (FAD) (Spaan et al, 2005). We observed a strong synthetic lethal interaction between SLC25A32 and known riboflavin transporter SLC52A2 (Fig. EV3C). SLC25A32's interactions with metabolic enzymes were similar to those of SLC25A51, a known mitochondrial nicotinamide dinucleotide NAD transporter (Luongo et al, 2020; Girardi et al, 2020a; Kory et al, 2020). Taken together, these interactions support that SLC25A32 transports FAD (Santoro et al, 2020; Peng et al, 2022), either exclusively, or in addition to THF.

In galactose, a much lower number of interactions (16) were scored in the metabolic core enzyme gene set (Fig. 3D). This is mostly due to our saturation filter removing synthetic lethality between two genes based on dLFC effect size. Grown in galactose, a higher number of mitochondrial SLCs and TCA cycle enzymes such as MPC1/2, PDHB, and CS, became essential as single KO (Fig. 3C), and accordingly the saturation filter removed 68 interactions compared to 33 in glucose. Notably, the pattern of genetic interactions within SLCs and TCA cycle enzymes reversed. In glucose, synthetic lethal interactions were observed between SLCs (MPC1/2, SLC25A1, SLC25A11) and enzymes (PDHB/A1, ACO2, CS) presumably because cells could reroute TCA cycle to compensate for the loss of one gene but not for two. In contrast, in galactose those genes became essential as single KO because cells must rely more on the TCA cycle for generation of biosynthetic precursors, and synthetic viable interactions were not scored as the second KO had no additional effect. Comparing genetic interactions in hypoxia, galactose, and antimycin with glucose, we found that 186 out of all 2357 gene pairs (8%) changed in at least one condition (Appendix Fig. S1).

The screen was additionally performed under glucose and hypoxia using Cas9 (Fig. EV4A and Dataset EV6, Cas9-SLCxEnzyme) and 47 interactions were detected in either the Cas12a or the Cas9 screens (Fig. EV4B). For some interactions, the direction of the interaction was reversed. For example, the interaction between MPC1 and MPC2 was detected in both the Cas12a- and Cas9-SLCxEnzyme screens but was classified as synthetic lethal in the Cas12a screen and synthetic viable in the Cas9 screen (Fig. EV4C). MPC1 and MPC2 must assemble as a heterodimer to form the functional mitochondrial pyruvate carrier (Bricker et al, 2012). Accordingly, as single knockouts (KOs), MPC1 and MPC2 showed nearly identical fitness phenotypes within each

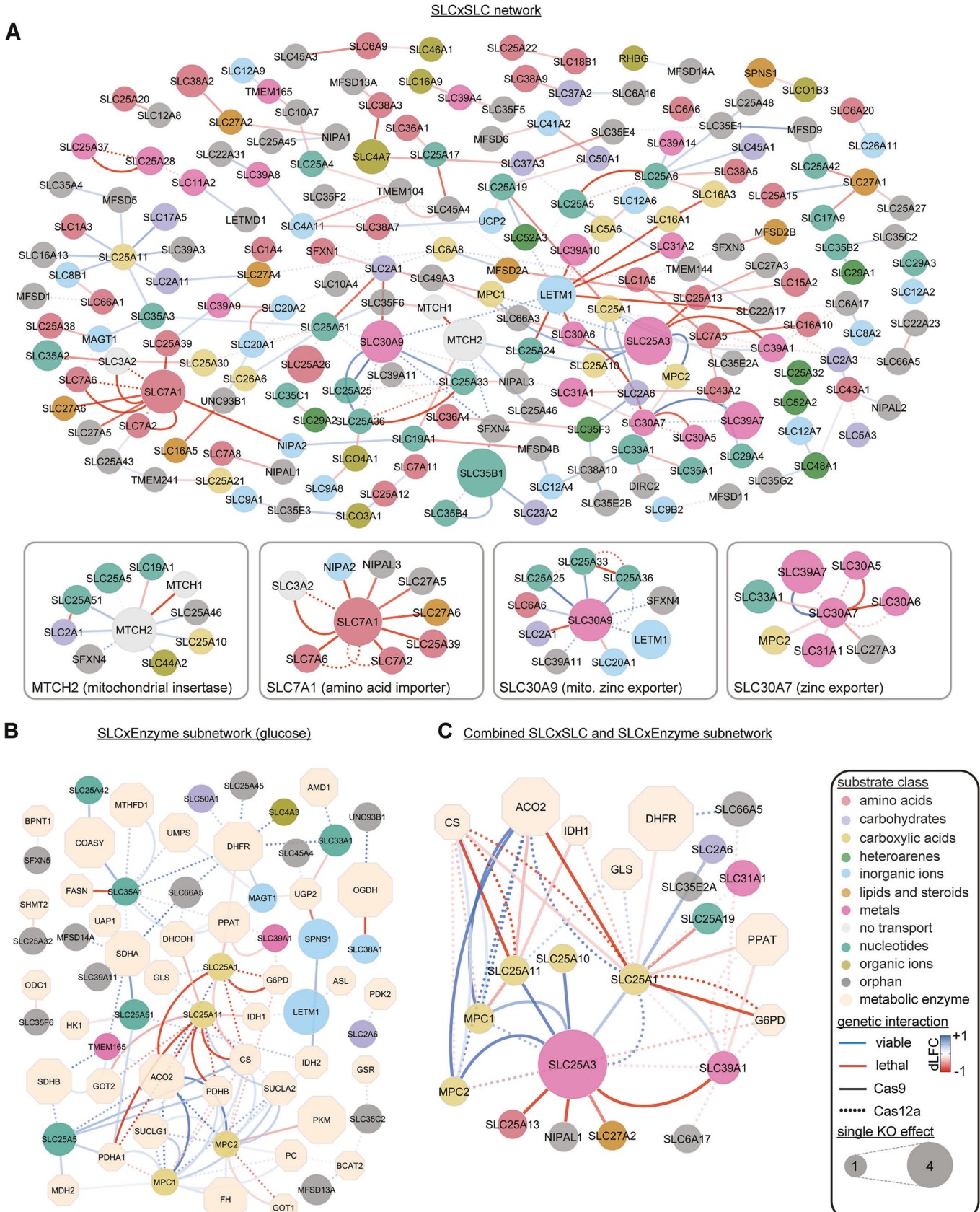

**Figure 4.  Combined network views of genetic interaction maps.**

(A) Network fusion of the Cas12a-SLCxSLC and Cas9-SLCxSLC screens showing 238 genetic interactions (edges) among 197 SLCs (nodes). For clarity, Cas12a interactions (padj < 0.1) involving six essential SLC were removed in the complete view, resulting in 69 edges. These are depicted together with 169 Cas9 edges, derived from the more conservative padj <0.005 cutoff. In contrast, the zoomed-in boxed views contains all unfiltered Cas12a (padj < 0.1) and Cas9 (padj < 0.01) interactions, highlighting subnetworks with functional enrichments, such as amino acid importers and zinc transporters. Different cutoffs were chosen solely to provide a manageable number of interactions for static illustration purposes. The full network is better viewed at www.re-solute.eu/resources/dashboards/genomics/network/ where interactive features (zooming, panning, filtering) facilitate easier navigation through this genetic interaction map. (B) Subnetwork selection of the combined Cas12a-SLCxEnzyme and Cas9-SLCxEnzyme genetic interactions in standard high-glucose medium. (C) Subnetwork selection of the combined genetic network, demonstrating strong interactions among mitochondrial carboxylate transporters, TCA cycle enzymes, and carbohydrate transporters. Shown are all SLCxSLC and SLCxEnzyme interactions involving SLC25A3, SLC25A1, and SLC39A1.

screen, but the effects were in opposite directions between the two systems. The single KO effects for MPC1 and MPC2 were modest, with LFC(week3) values of 0.60 and 0.56 (Cas12a) or −0.44 and −0.56 (Cas9). The observed value (LFC_obs) for the double KO was almost identical to the single KO effects, with values of 0.65 (Cas12a) and −0.40 (Cas9), and accordingly a genetic interaction was scored in both screens. Similar discrepancies were seen for other interactions that are closely related by metabolic pathways, such as MPC1-PDHA1, or SLC25A1-ACO2. These relatively mild differences in growth phenotype might be explained by clonal variations in between the HCT 116 clones used for both orthogonal screens. In conclusion, while our genetic interaction mapping approach is effective in identifying functional relationships, the direction of the interaction (lethal vs viable) should be carefully evaluated on a case-by-case basis.

Taken together, these screens demonstrate that genetic interactions between SLC transporters and metabolic enzymes can reveal insights into cellular metabolism under different conditions. Notably, the interactions within the mitochondrial transporters and TCA cycle enzymes indicate a high capacity for compensatory rerouting of metabolic fluxes, which breaks down when multiple genes are ablated. Additionally, the discovery of context-specific interactions, such as those involving SLC39A1 with one-carbon metabolism enzymes under hypoxic conditions, suggests that SLCs play a dynamic role in adapting cellular metabolism to environmental changes.

## Combined network analysis of genetic interactions

To gain deeper insights into functional relationships between SLCs, we conducted an analysis based on the broader network representation that combined genetic interactions from both Cas12a- and Cas9-mediated SLCxSLC screens. This allowed us to create a fused network which allowed us to detect functional SLC sub-networks (Fig. 4A). Manual examination of four sub-networks demonstrated that genetic interactions are capable of inferring SLC function based on common localization, substrate, or function (Fig. 4A).

MTCH2, an outer mitochondrial membrane (OMM) protein involved in the insertion of α-helical membrane proteins into the OMM (Guna et al, 2022), interacted with nine SLCs. Six of these SLCs are also localized in the mitochondria, underscoring MTCH2's critical role in mitochondrial function. Interaction partners included SLC25A51, a mitochondrial NAD transporter, the ADP/ATP exchanger SLC25A5, and MTCH1, which can partially compensate MTCH2 loss (Guna et al, 2022). Additionally, MTCH2 exhibited synthetic viable interactions with SLC25A46,

another OMM protein involved in mitochondrial fusion and fission (Schuettpelz et al, 2023; Janer et al, 2016). Other interactions with SLC44A2, a proposed mitochondrial choline importer (Bennett et al, 2020), and with the mitochondrial orphan transporter SFXN4 reinforced this mitochondrial sub-network around MTCH2.

SLC7A1 interacted with SLC3A2, which is required for the assembly of various SLC7 family transporters. Additionally, SLC7A1 interacted with other amino acid transporters, the magnesium transporter NIPA2 and the orphan transporter NIPAL3, suggesting a broader role in coordinating amino acid and ion transport. SLC7A1 also interacted with SLC27A6 and SLC27A5, both involved in long-chain fatty acid import, highlighting its central role in nutrient transport and a potential connection to lipid metabolism in HCT 116 cells.

A third cluster was centered around SLC30A9, which exhibited synthetic viable interactions with mitochondrial pyrimidine transporters SLC25A33 and SLC25A36, and the Mg-ATP transporter SLC25A25. Previous studies have shown that SLC25A25 can buffer the negative growth effects of SLC30A9 knockout in C. elegans by regulating mitochondrial zinc import (Ma et al, 2022). This suggests that SLC25A33 and SLC25A36 might also play roles in mitochondrial zinc homeostasis, potentially coordinating zinc levels with pyrimidine metabolism and ATP transport.

Additional synthetic lethal interactions were observed between two complexes that mediate zinc export from the cytosol through the Golgi apparatus and vesicular compartments. Specifically, interactions were observed between the zinc exporter homo-oligomer SLC30A7 and the hetero-oligomer SLC30A5/SLC30A6 (Suzuki et al, 2005), that are all expressed in the ER or in the Golgi apparatus. Moreover, the negative growth effect in SLC39A7 KO cells was partially mitigated by the simultaneous KO of SLC30A7, suggesting that intracellular zinc deficits resulting from the loss of the zinc importer SLC30A7 can be compensated by targeting the zinc exporter SLC39A7. These and similar findings could inform the development of inhibitors aimed at correcting zinc imbalances.

The joint network analysis of the Cas12a- and Cas9-SLCxEnzyme screen (Fig. 4B) also revealed a cluster of synthetic lethal interactions centered around the mitochondrial citrate carrier SLC25A1 and the oxoglutarate/malate carrier SLC25A11. Both showed strong synthetic lethal interactions with mitochondrial enzymes involved in pyruvate break-down (PDHA1, PDHB, CS), indicating that cells can compensate for the loss of these metabolic genes by rerouting metabolic flux away from the TCA cycle and through the cytosol. This was further supported by synthetic lethal interactions with cytosolic IDH1, pointing to citrate being converted to α-ketoglutarate in the cytosol and then transported back to mitochondria via SLC25A11 in exchange for malate.

Additionally, a strong interaction network was observed around MPC1/2 with large dLFC values, showing synthetic viable interactions with TCA cycle enzymes such as SUCLA2/G1 and SDHA/B. The loss of these TCA cycle enzymes disrupted the TCA cycle sufficiently to mask the effect of ablating MPC1/2.

When combining the SLCxSLC and SLCxEnzyme screens (Fig. 4C), we noticed that SLC39A1, a zinc transporter (Gaither and Eide, 2001), shared many interaction partners with SLC25A1, specifically SLC25A3, Glucose-6-Phosphate Dehydrogenase (G6PD) and Phosphoribosyl Pyrophosphate Amidotransferase (PPAT). A direct interaction between SLC25A1 and SLC39A1 was also observed in both Cas12a- and Cas9 SLCxSLC screens and both genes had similar single KO effects across different growth conditions (Fig. EV3C). Down-regulation of SLC39A1 expression has been associated with prostate cancer pathogenesis and is linked to poor prognosis (Franklin et al, 2005), with most studies focusing on zinc's direct effects on apoptosis or cell growth inhibition (Golovine et al, 2008; Feng et al, 2002; Liang et al, 1999). Consistent with these reports, we found that single KO of SLC39A1 increased the growth rate in HCT 116 cells (Fig. EV3C). However, its genetic interaction partners were unexpected, showing no interactions with other metal transporters but rather with SLCs and metabolic enzymes related to citrate metabolism (Fig. 4C). This prompted us to investigate whether the increased cell growth rate from SLC39A1 knockout is mediated through metabolic pathways rather than direct effects of zinc on apoptosis.

## Knockout of SLC39A1 rewires mitochondrial metabolism and triggers transcription of the NF-κB regulatory protein BCL3 and transcription factor STAT3

We first further examined the genetic interaction between SLC25A1 and SLC39A1, identified as synthetic lethal in the Cas12a-SLCxSLC and synthetic viable in the Cas9-SLCxSLC screen. Single KO effects in the Cas12a-SLCxEnzyme screen under glucose conditions showed that SLC39A1-KO and SLC25A1-KO sgRNAs were steadily enriched over time with single KO LFCs of 1.15 and 0.35 by week 5, respectively (Fig. EV5A). The relative growth advantage of both gene KOs was reversed under antimycin or galactose conditions but remained largely unaffected by hypoxia. In the Cas9-SLCxEnzyme screen on the other hand, SLC39A1-KO remained constant over time while SLC25A1-KO showed a similar pattern as in the Cas12-SLCxEnzyme screen (Fig. EV5B). These results demonstrate that SLC39A1-KO had significantly different growth impacts on the two HCT 116 clones, possibly explaining the opposite directionality of the SLC25A1-SLC39A1 interaction in both screens.

To understand the immediate effects of SLC39A1-KO, we utilized other RESOLUTE -omics data sets measuring the impact of inducible SLC39A1 expression on metabolites (Wiedmer et al, 2025), protein interactions (Frommelt et al, 2025), and transcripts.

First, we examined the immediate effects of SLC39A1 re-expression in a SLC39A1-KO background, analyzing metabolites in HCT116-SLC39A1-KO-OE cells after inducing expression with doxycycline (Fig. 5A). SLC39A1 re-expression led to significant changes in TCA cycle intermediates, including aconitic acid, citric acid, and α-ketoglutaric acid, as well as derivatives such as aspartic acid and malic acid. Additionally, notable changes in nucleotide levels were observed that are linked to mitochondrial function by

de novo purine synthesis pathways. This metabolic profile aligns with a role of SLC39A1-KO in mitochondrial function, possibly through zinc-dependent inhibition of mitochondrial aconitase, as previously reported for prostate cancer cells (Singh et al, 2006).

Next, we analyzed transcriptomics data from HCT116-SLC39A1-KO-OE and HCT116-SLC39A1-KO cell lines. For the KO-OE cell line, we first compared the RNA-seq LFC values after doxycycline induction for SLC39A1 and a control GFP-OE cell line, to filter out unspecific effects from protein overexpression and doxycycline induction. Transcripts with >0.5-fold change in the SLC39A1-KO-OE cell line and >0.5-fold difference from the GFP-OE cell line were considered differentially impacted by doxycycline-induced SLC39A1 expression (Fig. EV5C). Strikingly, 11/31 upregulated genes were associated with the ontology term *Hypoxia* in the MSigDB Hallmark 2020 database (padj = 1.29E−13, Odds Ratio = 57.6), supporting a role of SLC39A1 in OXPHOS.

Comparison of transcript LFC values between SLC39A1-KO and SLC39A1-KO-OE (Fig. 5B) revealed reversibly up- and down-regulated genes, including genes related to stress signaling (e.g., BCL3, STAT3) and cell cycle division (e.g., CDC6, CDC45). Gene set enrichment analysis indicated upregulation of terms related to inflammation and NF-κB signaling (Fig. 5C). While this transcriptional signature can explain the observed increase in cell growth rate, they do not provide a mechanistic explanation for how SLC39A1 KO is connected to these transcriptional changes.

Subcellular localization of SLC39A1 has been reported in different cellular compartments and may depend on cell type, including at the plasma membrane (Gaither and Eide 2001; Franklin et al, 2005), endoplasmic reticulum (Gaither and Eide, 2001), intracellular vesicles (Milon et al, 2001; Huang and Kirschke, 2007) and outer mitochondrial membrane (Cho et al, 2019). To investigate this further, we analyzed immunofluorescence co-localization data from HEK293 cells expressing HA-tagged SLC39A1 (HEK-SLC39A1-WTOE) (Goldmann et al, 2025). SLC39A1 showed a diffuse expression, not colocalizing with mitochondrial markers but partially with Golgi markers (Fig. 5D). We then turned to interaction proteomics (Frommelt et al, 2025), where affinity-purification mass-spectrometry (AP-MS) was performed by SLC39A1 pulldown in HEK293-SLC39A1-WTOE cells to identify direct protein interaction partners that might explain the observed metabolic and transcriptional changes. Interestingly, we found a protein–protein interaction (PPI) with SLC30A9, a mitochondrial zinc exporter (Fig. 5E) (Deng et al, 2021; Ma et al, 2022). A synthetic lethal interaction between SLC39A1 and SLC30A9 was also found in our Cas12a-SLCxSLC screen (Figs. 2A and EV2G). Gene ontology enrichment of all PPIs highlighted terms related to membrane protein trafficking and ER and Golgi membrane organization (Fig. EV5D), solidifying the multi-subcellular localization expression of SLC39A1.

Prompted by the genetic interaction between SLC39A1 and SLC25A1, we examined the genetic interaction network of SLC39A1. We observed profound and pleiotropic effects of SLC39A1 knockout, leading to increased cell growth rates and upregulation of anti-apoptotic genes. Given the extensive synthetic lethality network surrounding SLC39A1 (Fig. 4C), especially the observed synthetic lethality with SLC25A1, we hypothesized that the effects of SLC39A1 knockout were mediated by metabolic

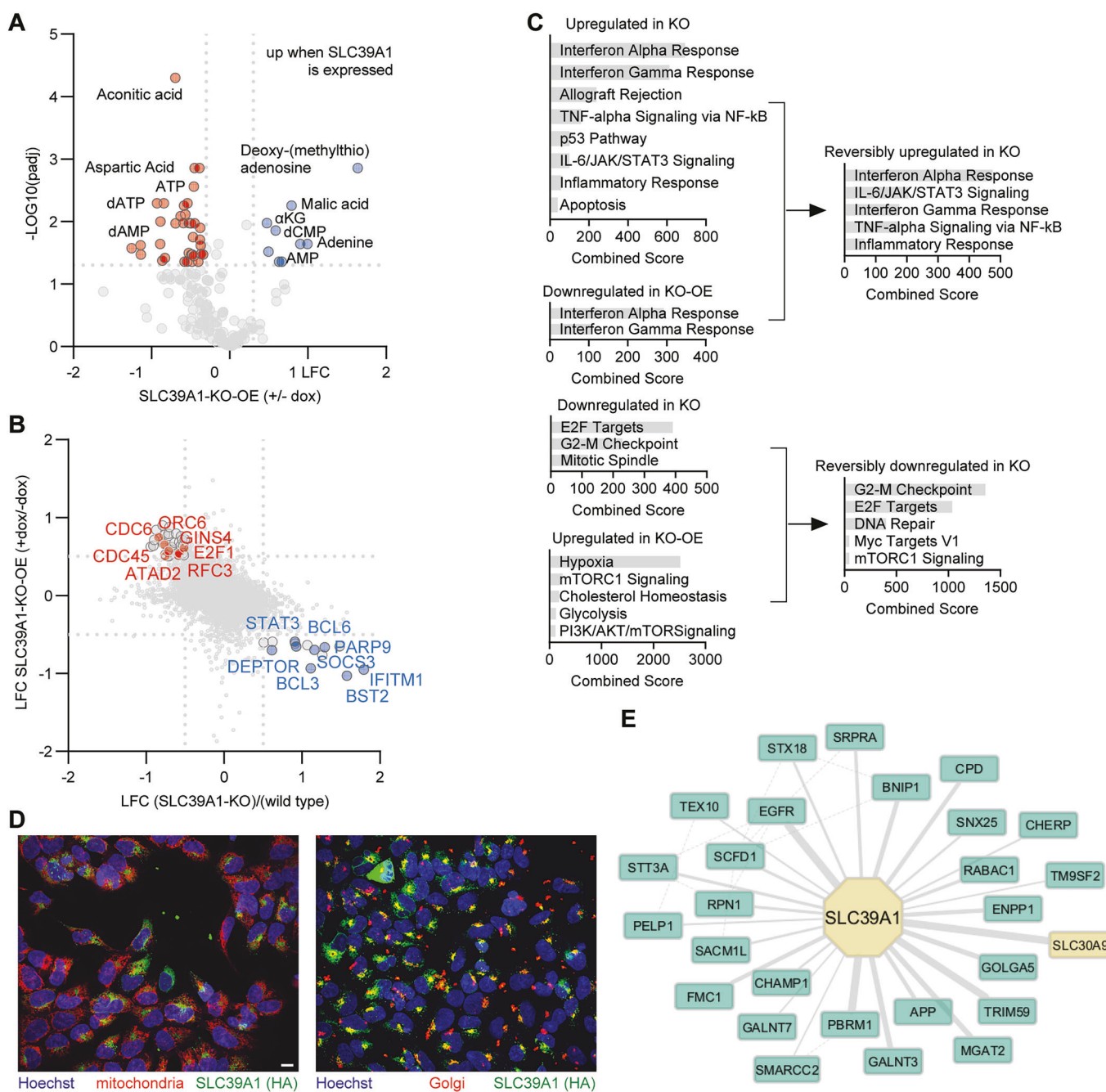

**Figure 5. Integrative analysis of other RESOLUTE data on SLC39A1.**

(A) Targeted metabolomics of HCT116-SLC39A1-KO-OE cells after doxycycline-induced re-expression of SLC39A1. Mean of four replicates. Significance was determined using two-way ANOVA and p values were adjusted using the Benjamini-Hochberg procedure. (B) Transcriptomics of HCT116-SLC39A1-KO-OE cells after doxycycline-induced re-expression of SLC39A1 and compared to HCT116-SLC39A1-KO. (C) EnrichR MSigDB gene set enrichment analysis of transcriptomics for HCT116-SLC39A1-KO, -KO-OE, and combined. (D) Immunofluorescence images of HEK293-SLC39A1-WTOE cells stained for mitochondria (left image, red, Mitotracker Orange CMTMRos) or Golgi (right image, red, anti-Giantin), nuclei (blue, Hoechst 33342), and SLC39A1 (green, anti-HA). Scale bar 5 μM. (E) AP-MS interaction proteomics data with SLC39A1-pulldown. Solid lines indicate protein-protein interactions (PPIs) with their weight describing the interaction probability score. Dashed lines are supplemented PPIs from the BioGrid database.

alterations. Future work will be dedicated to the further elucidation of the mechanistic link. In general, our data set warrants several dedicated studies to unravel the basis for many genetically defined functional relationships.

## Discussion

In his seminal work "The Logic of Scientific Discovery", Karl Popper emphasized that empirical testing forms the foundation of

scientific progress, best achieved without influence from prior theories and beliefs (Popper, 2005). Genetic screens epitomize this philosophy by enabling the systematic and comprehensive testing of genetic perturbations relatively free from bias. The resource we provide here adheres to this principle by testing the effects of combinatorial double knockouts (KOs) of all transporters with other SLCs and several metabolic enzymes under different growth conditions, assessing simple cellular fitness. This work should advance previous studies on genetic interactions among transporters in five key aspects: First, the scope is unprecedented for SLC transporters and probably any other human group of genes (Girardi et al, 2020b; Shi et al, 2022). Second, we performed all single KOs and double KOs simultaneously, avoiding the need for sequential single KOs that require subculturing, are very labor intensive and carry the risk genetic compensation and drift over time (Girardi et al, 2020b, 2020a). Third, we paired SLCs with metabolic enzymes to elucidate how SLC transporters interact with metabolic pathways under different growth conditions. Fourth, we provide an open-access highly interactive and user-friendly dashboard for data exploration and hypothesis generation (https://re-solute.eu/resources/dashboards/genomics). Fifth, it offers part of the functional genomics backbone for the interpretation of data obtained with interaction proteomics, metabolomic and knowledge scouting and integration of the three other RESOLUTE studies.

To identify genetic interactions between two genes, we quantified the growth phenotype of double KOs and scored them whenever the double KO effect was stronger or weaker than the combined single KO effects. This approach has some inherent limitations that are important to consider when interpreting this data set. First, the definition of the expected double KO effect (LFC_exp) significantly impacts the detected genetic interactions. Various metrics exist, and we defined LFC_exp as the sum of LFC_KO1 and LFC_KO2. LFC_exp is then compared to the measured double KO effect (LFC_obs) (Mani et al, 2008). As such, the scoring metric dLFC, representing the difference between LFC_obs and LFC_exp, is derived from three different experimental measurements, potentially introducing noise and error. Since we are testing significance conservatively, it is likely that this noise leads to an underestimation of the true number of genetic interactions. Secondly, growth phenotype is a readout with several limitations. While it provides a convenient and scalable quantitative measure, it requires culturing cells over several weeks to amplify small changes in growth rates. Moreover, the readout is physically constrained since both cell death and growth arrest effects can be saturated when depleting two genes with negative growth phenotypes; in other words, a dead cell can hardly be further slowed down in growth. Similarly, the effect of two genes that improve cell growth might not add up as expected since there are other biochemical and biophysical constraints for the maximal growth rate of a cell, such as nutrient availability or DNA replication speed. This is likely to severely limit our assay's capability to capture genetic interactions of genes that have strong single KO growth phenotypes. Finally, interpreting the nature of genetic interactions detected in proliferation-based screens can be challenging. Interactions driven by a shared substrate, as seen with SLC30A5-SLC30A7 (zinc), SLC25A33-SLC25A36 (pyrimidine nucleotides), or SLC20A1-SLC20A2 (phosphate), can easily be explained by functional redundancy of two genes associated with the same transport function. Alternatively, two genes might interact

physically and assemble to form a functional unit, as in the case of MPC1-MPC2 (Bricker et al, 2012). where the KO of one gene is sufficient to cause the phenotype, and the effect of the second KO is masked. Interactions can also be indirect and non-local, for instance among genes such as SLC2A1 and SLC25A51 involved in the same pathway, here in glucose energy metabolism.

However, interactions can also arise between genes that lack a common substrate or an obvious functional relationship, as many factors and pathways indirectly affect growth rate (Bellay et al, 2011). While this highlights the usefulness of such screens in providing a hypothesis-free functional mapping, the challenge lies in understanding the mechanisms underlying the interaction network. Notably, each genetic interaction in a network of an individual gene may be driven by a different mechanism. This mechanistic diversity, while informative and multifaceted, requires careful interpretation and can complicate efforts to cohesively describe the function of an individual uncharacterized gene. For instance, we found that genes with pleiotropic effects, such as those causing growth arrest, inducing of apoptosis, or perturbing energy metabolism, often genetically interacted with many genes without a distinguishable functional association. LETM1, an essential mitochondrial calcium transporter (Shao et al, 2016), showed six interactions with SLCs of different substrate classes and localizations. Readouts beyond proliferation, such as surface marker expression, sensors detecting individual metabolites, or reporter genes, hold potential for identifying interactions between SLCs sharing functions without growth phenotypes. However, these approaches come with their own set of trade-offs, typically offering a narrower viewpoint and having lower scalability, for instance due to the need for sample enrichment through cell sorting.

Our results show that combinatorial double KO screens when using the same library, CRISPR enzyme, and cell clone, are technically very replicable. However, they will result in a unique set of interactions that depends both on clonality and growth conditions, even when using the same cell line. Consequently, the map presented here represents one of many possible genetic interaction landscapes in a human cell line. Despite these considerations, our proliferation-based screen maps the genetic interactions of the SLC superfamily in the context of a very fundamental phenotype, that is cell fitness. Given the stringent criteria used for scoring interactions, this map provides a core backbone of functional relationships that should stand the test of time, representing a holistic functional annotation by genetics for many transporter genes. The relatively low rate of detecting essential SLCs and synthetic lethality between SLC pairs suggests a high adaptability of cells to single and double knockouts, at least when assessing growth phenotypes in standard media. This adaptability aligns with the known metabolic plasticity observed in cancer cells (Fendt et al, 2020). Beyond cancer cells, this plasticity is important for cellular resilience to changing environments and was theorized to have been a key factor in the origin of life (Seebacher and Beaman, 2022). These findings also echo the well-documented context-dependence of gene essentiality in yeast (Hillenmeyer et al, 2008), highlighting a common theme across very different organisms.

Specifically, we detected only 437 (1.3%) genetic interactions among the possible 33,153 SLCxSLC combinations. Notably, 15/69 (22%) of Cas12a- and 21/169 (12%) Cas9-SLCxSLC interactions were observed among mitochondrial SLCs (expected by chance:

5.7%). Several factors might explain the scarcity of strong synthetic lethal interactions for non-mitochondrial SLCs. First, many essential transport functions at the plasma membrane may be carried out by more than two SLCs, enhancing cellular robustness to changing environment, such as fluctuating metabolite concentrations that are not under homeostatic control. This is not necessary for mitochondrial carriers, because the cytosol is a protected environment. Second, redundancy likely extends beyond commonly transported substrates. When cultured in non-physiological growth media that is supra-physiologically rich in nutrients and other vital compounds, the loss of transport activity for a specific substrate may be compensated by endogenous synthesis of the missing substrate or by switching to alternative metabolic pathways that buffer this loss (Rossiter et al, 2021). While we have performed targeted metabolism on single knockout-overexpression cells for each individual SLC (Wiedmer et al, 2025), it would now be interesting to measure metabolites in many interesting genetic interaction cases, something that regretfully, is beyond the scope of this study.

By comparison, we detected a relatively high percentage (2.5% for Cas12a-SLCxEnzyme in glucose) of often strong genetic interactions between SLCs and metabolic enzymes. Like the SLCxSLC interactions, those were more frequently observed between mitochondrial SLCs and enzymes driving the TCA cycle. This suggests that pairing SLCs with metabolic enzymes could yield more informative results than screening SLCs alone, guiding the design of more effective future studies. The cost-effectiveness, flexibility and efficiency of combinatorial KO screens demonstrated by others and in this study (DeWeirdt, 2020; Li et al, 2022), opens the opportunity to investigate genetic dependencies for transporters and all proteins in many more functional contexts.

Beyond hypothesis generation related to SLC transporter function, this data set may be of practical use. First, it directly offers researchers a set of combinatorial genetic perturbations to test for their gene of interest. Secondly, these novel genetic interactions can also be exploited for correcting diseases caused by SLC loss-of-function. This is important, because to date a loss-of-function of a transporter is difficult to correct using drugs. Positive allosteric modulators for SLC transporters have been very difficult to identify, with a first example for the excitatory amino acid transporter EAAT2 (Kortagere et al, 2018) later being questioned (van Veggel et al, 2024). Therefore, a chemical correction of a loss-of-function mutation like for some alleles of the ABC transporter CFTR has not been achieved for SLCs (Quon and Rowe, 2016). However, correcting disease phenotypes due to loss-of-function mutation in one SLC might be achieved by chemical inhibition of a second SLC that exhibits a genetic interaction, such as correcting a defect in a zinc importer by inhibiting a zinc exporter. This approach could also be effective at the organismal level, as seen in the treatment of diabetes by blocking glucose reuptake in the kidney through SGLT2 inhibition with gliflozins. Thirdly, such genetic interactions can also be exploited for assay development using cell survival as readout, essentially mimicking a genetic interaction by chemical inhibition. This methodology is termed paralog-dependent isogenic cell assay (PARADISO) and has been successful in developing specific SLC16A3 inhibitors (Dvorak et al, 2023). PARADISO might be especially important for otherwise intractable targets for assay development such as mitochondrial carriers (Dvorak et al, 2021). Of particular interest among the identified genetic interactions is the relationship between SLC39A1 and SLC25A1. While we were unable to fully elucidate the mechanism behind the positive effects of SLC39A1 knockout on cell growth, the downregulation of SLC39A1 has been observed in prostate cancer (Franklin et al, 2005). Our discovery of significant interactions between SLC39A1 and genes related to citrate metabolism, notably SLC25A1, suggests potential new therapeutic targets for further investigation in the context of prostate cancer and other SLC39A1-related disorders.

Overall, our genetic interaction map of SLC transporters presents a cohesive functional assessment for a large group of human genes. This interaction map should prove to be a valuable resource for the scientific community and contribute to the expectation raised in preparation of the RESOLUTE effort (César-Razquin et al, 2015; Superti-Furga et al, 2020). The full data set is freely available to the scientific community at https://re-solute.eu/resources/dashboards/genomics/. We hope it will generate hypotheses that can additionally strengthened using the other complementary data sets provided, justifying dedicated efforts to elucidate the molecular mechanisms of the functional links reported here. We also expect this will extend beyond individual interactions, offering insights into the broader network of SLC transporters and their connections to metabolism, cellular homeostasis, and disease.

# Methods

**Reagents and tools table**

| Reagent/Resource | Reference or Source | Identifier or Catalog Number |
|---|---|---|
| **Experimental models** | | |
| HCT 116 | ATCC | Cat# CCL-247; RRID: CVCL_0291 |
| **Recombinant DNA** | | |
| Combinatorial Cas12a and Cas9 KO libraries | This paper | N/A |
| pRD_052 | DeWeirdt et al | Addgene plasmid, Cat# 136474 |
| pRD_174 | DeWeirdt et al | Addgene plasmid, Cat# 136476 |
| lentiCas9-Blast | Sanjana et al | Addgene plasmid, Cat# 52962 |
| pWRS1001 | Li et al | Addgene plasmid, Cat# 192205 |
| **Antibodies** | | |
| Anti-HA | Novus | #NBP1-91938 |
| Anti-Giantin | Novus | #NB600-362 |
| **Oligonucleotides and other sequence-based reagents** | | |
| sgRNA sequences | This paper | Tables EV1, EV2, EV3, EV5, EV6 |
| Cas12a library amplification primer | DeWeirdt et al and this paper | Table EV7 |
| Cas12a library sequencing primer | DeWeirdt et al | Table EV7 |
| Cas9 library amplification primer | This paper | Table EV7 |
| Cas9 library barcoding primer | Veeranagouda et al | Table EV7 |
| **Chemicals, Enzymes and other reagents** | | |

| Reagent/Resource | Reference or Source | Identifier or Catalog Number |
|---|---|---|
| Nystatin | Merck | #N3503-5MU |
| Fetal bovine serum | Gibco | #10270-106 |
| Penicillin-Streptomycin | Gibco | #15140-122 |
| Blasticidin | Invivogen | #ant-bl-5b |
| Puromycin | Sigma | #P8833-100MG |
| Endura Electrocompetent cells | Biosearch Technology | #60242-2 |
| Q5 DNA polymerase | NEB | #M0491L |
| RPMI | Sigma | #R8758 |
| QIAamp DNA Mini Kit | Qiagen | #51304 |
| QIAGEN Genomic-tip 100/G | Qiagen | #10243 |
| QIAGEN Genomic-tip 500/G | Qiagen | #10262 |
| PacI enzyme | NEB | #R0547S |
| XbaI enzyme | NEB | #R0145S |
| Phusion HF buffer | NEB | #B0518S |
| Pfu-Sso7d polymerase | Self-purified | https://barricklab.org/twiki/bin/view/Lab/ProtocolsReagentsPfuSso7d |
| RNeasy 96 Kit | Qiagen | #74181 |
| NEBNext Ultra II Directional RNA Library Prep Kit | NEB | #E7760 |
| NEBNext Poly(A) mRNA Magnetic Isolation Module | NEB | #E7490 |
| NEBNext Multiplex Oligos for Illumina | NEB | #E7600 |
| AmpliClean Magnetic Beads | Nimagen | #AP-050 |
| BsmBI-v2 | NEB | #R0739S |
| BbsI-HF | NEB | #R3539S |
| T4 ligase | NEB | #M0202S |
| Mitotracker Orange CMTMRos | Thermo | #M7510 |
| **Software** | | |
| Cutadapt | Martin et al | https://github.com/marcelm/cutadapt/blob/main/doc/guide.rst |
| MAGeCK | Li et al | https://github.com/liulab-dfci/MAGeCK |
| DESeq2 | Love et al | https://github.com/thelovelab/DESeq2 |
| Enrichr | Xie et al | https://maayanlab.cloud/Enrichr/ |
| Galaxy platform | The Galaxy community | https://usegalaxy.org |
| GraphPad Prism | GraphPad Software Inc | https://graphpad.com |
| **Other** | | |

## Cell lines

The HCT 116 (RRID:CVCL_0291) cell line was purchased from ATCC (CCL-247™) and cultured in RPMI (#R8758 Sigma) supplemented with 10% Fetal Bovine Serum (10270-106, Lot 42F8381K, Gibco) and Penicillin-Streptomycin (15140-122, Gibco). All cell lines were grown in standard cell culture dishes at 37 °C with 5% $CO_2$. To generate enCas12a and Cas9 expressing clones, HCT 116 cells were stably transduced with lentiviral vectors pRDA_174 (Addgene #136476) or lentiCas9-Blast (Addgene #52962), respectively. After five days of selection with blasticidin, single-cell clones were prepared by cell sorting and subsequent expansion. A total of eight clones each were tested for enCas12a and Cas9 activity using a CD46/CD81 combinatorial KO assay. The clones with the highest double KO efficiencies (HCT116-enCas12a-c8 and HCT116-Cas9-p2A-Blast-c1) were used for all subsequent pooled screens. Cell pools transduced with Cas12a-SLCxSLC sub-libraries were cultured in the presence of 100 U/ml Nystatin (Merck, #N3503-5MU).

## Combinatorial KO library design

For the Cas12a metal SLC transporter-focused library (Cas12a-Metal-SLCxSLC), 21 metal transporters expressed in HCT 116 (>1 transcript per million) were selected as targets. Three olfactory receptors (ORs) were added as negative controls to determine the effect of single SLC KOs. For each target, three enAsCas12a sgRNAs were designed per gene using CRISPick (Broad Institute) (DeWeirdt, 2020; Kim et al, 2018). All 72 sgRNAs were paired with each other in both orientations resulting in a total of 5184 combinations.

For the Cas12a-Control library, three sgRNAs each were designed for six commonly essential genes, each sgRNA paired with a sgRNA targeting an OR. As positive controls for synthetic lethal interactions, we selected six gene pairs that previously displayed a synthetic lethal phenotype (DeWeirdt, 2020). Three sgRNAs per gene were used in all 18 possible combinations for each of the six gene pairs. To monitor single gene KO, these sgRNAs were also paired with 6 sgRNAs targeting three different ORs.

Cas12a oligos for the Cas12a-Metal-SLCxSLC, Cas12a-control, and Cas12a-Metal-SLCxEnzyme libraries were synthesized as 134 nt ssDNA oligo pool, with two 23 nt Cas12a sgRNAs per oligo separated by a 20 nt direct repeat (DR) sequence (TAATTTCTACTGTCGTAGAT) as described previously (DeWeirdt, 2020), flanked by BsmBI restriction sites and PCR primer sequences for amplification and cloning. The libraries were ordered as one single-stranded DNA oligo pool (Twist) with different primer sequences for metal transporter and control library to allow separate amplification of each of the sub-libraries.

For the Cas12a-SLCxSLC library, three sgRNAs for each of the 258 SLCs expressed in HCT 116 were designed using CRISPick. These include the five members of the SLC66 family which were not originally targets of RESOLUTE and an uncharacterized trans-membrane protein (TMEM144) that showed structural similarity to SLCs. As controls, we chose 12 sgRNAs targeting 12 different ORs. Altogether, this resulted in 617,796 possible sgRNA combinations. Oligos for the Cas12a-SLCxSLC library were ordered as 786 FW and 786 RV ssDNA oligos (Twist), each containing a primer binding sequence, BsmBI site, one Cas12a sgRNA sequence and the DR sequence. The sequences on the RV oligos were reverse

complemented. To allow amplification of smaller sub-libraries, the FW oligos contained four distinct primer binding sequences (64 to 65 targets per primer). FW oligos with negative control sgRNAs were designed with all four primer binding sites to ensure that all SLC sgRNAs are paired with control sgRNAs in both orientations.

For the Cas9-control library, we selected the same six gene pairs as for the Cas12 control library Six sgRNAs per gene were designed using the VBC sgRNA design tool (Michlits et al, 2020) and combined in 12 combinations for each of the six gene pairs and each sgRNA was paired with 3 sgRNAs targeting ORs. Oligos were synthesized as 129–135 nt ssDNA oligo pools (Twist).

For the Cas9-SLCxSLC library, six sgRNAs were designed for each of the 258 expressed SLCs using the VBC sgRNA design tool (Michlits et al, 2020). As controls, we used 12 sgRNAs against 12 ORs. Oligos were ordered as 1560 FW and 1560 RV ssDNA oligos (Twist), each containing a primer binding sequence, BsmBI site, one Cas9 sgRNA sequence and an overlapping region with BpsI sites (GTTTCAGTCTTCCGGCGAAGACACAAAC) that was used for oligo recombination and insertion of VCR1-WCR3 tracrRNAs (Li et al, 2022). The sequences on the RV oligos were reverse complemented. To reduce the number of oligo combinations, we designed six different primer sequences for the six sgRNAs which allowed us to amplify six subsets of the library, each containing 12 sgRNA combinations per gene.

For the Cas9-SLCxEnzyme library, sgRNAs against the same synthetic lethality pairs as for the Cas12 SLC vs enzyme library were designed using the VBC sgRNA design tool (Michlits et al, 2020) and synthesized as 129–135 nt ssDNA oligo pool (Twist).

For the Cas12a SLC vs metabolic Enzymes library we designed 6 sgRNAs per gene and combined SLC genes with metabolic enzyme genes to get six sgRNA pairs for each gene pair. Each sgRNA was paired with three OR targeting sgRNAs as controls. The library consisted of four subgroups: (1) TCA cycle (9 SLCs vs 17 enzymes), (2) one carbon metabolism (8 SLCs vs 10 enzymes), (3) CoA metabolism (13 SLCs vs 14 enzymes) and core metabolism (49 SLCs vs 40 enzymes). Detailed information about library composition can be found in Dataset EV5 and EV6.

## Library cloning

The $2 \times 786$ nt ssDNA oligos for the Cas12a-SLCx SLC library were recombined using their common DR sequences in a PCR for 5 cycles without primer and 55 °C annealing temperature, followed by 16 cycles in four different reactions with added primer and the corresponding annealing temperatures for each primer pair. This resulted in four sub-libraries, each containing approximately 160,000 sgRNA combinations (64 to 65 SLCs and negative controls on the first sgRNA position and 258 SLCs and controls on the second position).

The 134 nt ssDNA oligos for the Cas12a control, metal SLC vs metal SLC and SLC vs metabolic enzymes libraries were directly amplified by PCR.

Amplified dsDNA oligos were purified using 1.8 x volume AmpliClean beads (Nimagen) and inserted between the BsmBI restriction sites of pRDA_052 (Addgene #136474) via golden gate cloning. For the SLC superfamily-wide library, we introduced two PacI restriction sites (one directly upstream of the U6 promoter, one 86 bp upstream of the EF1a promoter) into the pRD_052 vector before inserting the sgRNA library. Cloning reactions were

transformed into Stbl3 competent cells via heat shock transformation or to Lucigen Endura electrocompetent cells via electroporation to reach a coverage of at least 40 clones per oligo.

The $2 \times 1560$ ssDNA oligos for the Cas9 SLC vs SLC library were used in a PCR for 5 cycles without primer and 55 °C annealing temperature to combine the FW and RV oligos via their common overlap sequence, followed by 25 cycles in six different reactions with added primer and the corresponding annealing temperatures for each primer pair.

The resulting six sub-libraries were inserted into the BsmBI cloning site of the pWRS1001 vector (Addgene #192205) via golden gate cloning. The 129–135 nt ssDNA oligos for the Cas9 Control and SLC vs metabolic enzymes libraries were directly amplified by PCR and cloned into the pWRS1001 vector. All library cloning PCRs were performed using Q5 DNA polymerase (NEB).

Cloning reactions were then purified and digested with BsmBI to minimize empty backbone vectors and used for electroporation into Lucigen Endura electrocompetent cells (>40x coverage). To insert tracrRNAs between the Cas9 sgRNAs, the 250 bp VCR1-WCR3 sequence (Li et al, 2022) was synthesized as dsDNA fragment (Twist), subcloned into pDONR221, PCR amplified and cloned between the BbsI sites of each of the six sub-libraries via golden gate cloning. Cloning reactions were used for electroporation into Lucigen Endura electrocompetent cells (>40x coverage).

## Combinatorial KO screens

Lentiviral particles were produced in HEK293T cells and used to stably transduce enCas12a (HCT116-enCas12a-c8) or Cas9 (HCT116-Cas9-p2A-Blast-c1) expressing clones at an MOI of ~0.5. For the Cas12a-Metal-SLCxSLC screen, the library was mixed with the Cas12a-control library and transduced into $2 \times 10^7$ cells per replicate in triplicates to achieve a nearly 2000-fold coverage of the oligos. Cells were selected with puromycin for five days and afterwards passaged every 2–4 days for a total of 5 weeks. For each replicate, $2 \times 10^7$ cells were harvested for sequencing at week 1, 4, and 5 after transduction.

For the Cas12a-SLCxSLC screen, four independent sub-screens were performed with each of the four sub-libraries in triplicates. The control library was added to each of the sub-libraries before virus production. $7 \times 10^8$ cells were transduced per replicate and, after puromycin selection, $2.2 \times 10^8$ cells (corresponding to a ~1350x coverage) were passaged for a total of 5 weeks. For each replicate, $2.2 \times 10^8$ cells were harvested for sequencing at week 1, 2, 3, and 5 after transduction.

The Cas9-SLCxSLC library (mixed with the Cas9-control library) was transduced into $7 \times 10^8$ cells in duplicates and, after puromycin selection, $2.2 \times 10^8$ cells (~500x coverage) were passaged for 4 weeks. For each replicate, $2.2 \times 10^8$ cells were harvested for sequencing at week 1, 2, 3, and 4 after transduction. One of the two replicates was only cultivated until week 3.

Cas12a- and Cas9-SLCxEnzyme libraries were transduced into $9 \times 10^7$ cells in triplicates and $1.7 \times 10^7$ puromycin selected cells were passaged for 5 (Cas12a) or 6 (Cas9) weeks. Cas9 pools were cultivated in standard media and in hypoxia conditions (1% oxygen starting at week 2). Cas12a pools were cultivated in four conditions: (1) standard media; (2) hypoxia (1% oxygen); (3) 50 mM antimycin A; (4) low glucose media with galactose replacement (week 1–3: 20%, week 3–4: 0%, week 4–5: 10%). Treatments 2–4 were started

after one week under puromycin selection and two days recovery in standard medium (one day after week 1 samples were harvested). For each replicate, $1.7 \times 10^7$ cells were harvested for sequencing at week 1, 3, and 5 (Cas12a) or week 1, 2, 3, 4, and 6 (Cas9) after transduction.

Detailed protocols for the combinatorial Cas12a and Cas9 screens are available on Zenodo (#10354692, #12819677).

## Library amplification

For the Cas12a-Metal-SLCxSLC, Cas12a- and Cas9-SLCxEnzymes, and Cas12a- and Cas9-SLCxSLC, gDNA was purified from $2 \times 10^7$ (QIAamp DNA Mini Kit), $1.7 \times 10^7$ (QIAGEN Genomic-tip 100/G) or $2.2 \times 10^8$ cells (QIAGEN Genomic-tip 500/G), respectively. gDNA from Cas12a- and Cas9-SLCxSLC screens was digested with restriction enzymes to improve PCR amplification of sgRNA regions. Briefly, up to 1.5 mg gDNA was digested with 200 units PacI (Cas12a screen) or PacI and XbaI (Cas9 screen) (all NEB) for 48 h.

sgRNA sequences from Cas12a library gDNA samples were amplified with barcoded FW and RV primer that introduce Illumina adapter and read1 sequencing primer binding sites. Up to 1.2 mg gDNA was used as template in 12 ml volume (divided in $96 \times 125$ µl PCRs) in the following program: 98 °C for 30 s; 5 cycles: 98 °C for 10 s, 61 °C for 30 s, 72 °C for 30 s; 19 cycles: 98 °C for 10 s, 72 °C for 1 min; 72 °C for 2 min. For the samples of the metal transporter and SLC superfamily-wide sub-screen 1, Q5 Polymerase (NEB) was used, SLC vs SLC sub-screen 2–4 and SLC vs metabolic enzymes samples were processed with self-purified Pfu-Sso7d polymerase (Wang et al, 2004) and Phusion HF buffer (NEB). Primer sequences are shown in Dataset EV7. PCR products we double size selected with DNA purification bads and multiplexed for Illumina sequencing.

sgRNA sequences from Cas9 library gDNA samples were first amplified with staggered FW and RV primer that contain partial Illumina adapter sequences at the 5′ end using the same PCR program as above. Up to 1.2 mg gDNA was used as template (125 µg/125 µl volume PCR) and amplified with self-purified Pfu-Sso7d polymerase. An aliquot of the pooled amplified PCR products was double size selected with DNA purification bads and purified fragments were barcoded in a second PCRs (98 °C for 30 s; 6–8 cycles: 98 °C for 10 s, 61 °C for 30 s, 72 °C for 30 s; 72 °C for 2 min). PCR products we double size selected with DNA purification bads and multiplexed for Illumina sequencing.

## Illumina sequencing and read processing

Samples from the Cas12a-Metal-SLCxSLC screen were sequenced on a Hiseq3000/4000 machine, all other samples on a NovaSeq machine. Cas12a samples were sequenced as 100 bp single-end reads with a custom sequencing primer as described previously (DeWeirdt, 2020).

Cas9 samples were sequenced as $2 \times 65$ bp paired-end reads with standard Illumina sequencing primer. Read1 and read2 contained sequences of sgRNA1 and sgRNA1, respectively. Read1 reads were trimmed using cutadapt (Martin, 2011) to remove up- and downstream flanking sequences. The two sgRNAs from read1 and read2 were then joined using FASTQ Joiner (Blankenberg et al, 2010) before mapping to the reference sequences using MAGeCK.

## Data analysis for combinatorial KO screens

Due to lower sequencing quality, reads from the Cas12a-Metal-SLCxSLC screen were mapped to the reference sgRNA combinations using BWA (Li and Durbin, 2010) allowing 2-3 mismatches per read. Reads from all other screens were mapped to the reference sgRNA sequences using MAGeCK count (Li et al, 2014) allowing only perfect sequence matches.

For each sample, raw read counts were normalized to total mapped reads and gDNA samples from different time points were normalized to input samples (plasmid) to calculate the observed log2 fold changes (LFC_obs). The double OR targeting sgRNA combinations were used for normalization and for generating the null distribution of the log2 fold changes (LFC) of the individual oligos.

To determine the expected effect (LFC_exp) of each combinatorial sgRNA pair, the average LFCs of each of the two sgRNAs paired with the OR targeting sgRNAs in the corresponding orientations were summed up. The LFC_exp values were then normalized to generate the null distribution of the delta LFC_exp - LFC_obs (dLFC) values.

To confirm replicate reproducibility (Billmann et al, 2023), Pearson Correlation Coefficients (PCCs) were calculated between replicates on raw read counts, for double KOs at gene-level on LFC_obs and for the sum of single KOs at gene level LFC_exp (Appendix Fig. S2).

For each gene pair, the significance of a potential genetic interaction was determined using Student's paired t-Tests with a two-tailed distribution in Microsoft Excel. For this, we compared the LFC_obs and LFC_exp values derived from all sgRNA pairs targeting the corresponding gene pair, as described above. Due to the different library designs and number of sgRNA oligos per gene pair, we used varying scoring approaches for the five performed combinatorial screens. Briefly, LFC_obs and LFC_exp values from replicates and time points were either scored separately or samples were merged for higher statistical power (Table EV1). *P* values were adjusted using the two-stage linear step-up procedure of Benjamini, Krieger, and Yekutieli with GraphPad Prism. Only significant synthetic lethal interactions with a dLFC $< -0.3$ and synthetic viable interactions with a dLFC $> 0.3$ were scored.

To rule out interactions as the result of saturation effects that become apparent when adding the single KO effects of two essential interaction partners, synthetic viable interactions with LFC_exp values below $-0.3$ were excluded when the LFC_obs was lower than each of the single gene KO LFC values. Further, synthetic lethal and viable interactions for which the dLFC did not change between week 1 and week 3 by at least LFC 0.1-fold were excluded since we could not exclude that the changes in sgRNA abundances were caused by varying transduction efficiencies. Raw sequencing data are available at the Gene Expression Omnibus under the accession GSE269905.

## Targeted metabolomics

Cells were processed for targeted metabolomics as described in (Wiedmer et al, 2025). Briefly, $1.5 \times 10^5$ cells per well were seeded in quadruplicates in 24-well culture dishes in the absence or presence of 1 µg/ml doxycycline. Twenty-four hours after seeding, cells were washed and ice cold 80:20 MeOH:$H_2O$ containing a mixture of

isotopically labeled internal standards was added. Cells were then harvested by scraping and snap-frozen in liquid nitrogen. After thawing on ice, supernatant containing metabolites was clarified by centrifugation and used for LC-MS/MS analysis on a 1290 Infinity II UHPLC system (Agilent Technologies).

## Transcriptional analysis

Cells were processed for RNA sequencing as described in (Wiedmer et al, 2025). Briefly, $2.5 \times 10^5$ cells were seeded per well in duplicates in 12-well culture dishes in the absence or presence of 1 μg/ml doxycycline. Cell lines were harvested 24 h after doxycycline induction and RNA was purified using a RNeasy 96 Kit (Qiagen). RNA-seq libraries were prepared using the NEBNext Ultra II Directional RNA Library Prep Kit for Illumina #E7760, together with the NEBNext Poly(A) mRNA Magnetic Isolation Module #E7490 upstream and the NEBNext Multiplex Oligos for Illumina #E7600 downstream (all New England Biolabs). All libraries were sequenced as 50 bp paired-end reads on a NovaSeq 6000 Sequencing System (Illumina) and mapped to reference genome (Ensembl 98) using STAR (Dobin et al, 2013). Differential gene expression was determined using DESeq2 (Love et al, 2014). Gene set enrichment was performed using the Enrichr platform (KEGG 2021 Human) (Xie et al, 2021).

## Immunofluorescence

Imaging was performed according to the RESOLUTE expression analysis and co-localization protocol (Zenodo #7457346; https://zenodo.org/records/7457346).

# Data availability

Raw and processed sequencing data from the genetic interaction screens are available at the Gene Expression Omnibus under accession number GSE269905. Transcriptomics data for knockout and knockout-overexpression cell lines can be accessed at the RESOLUTE web portal (https://re-solute.eu/resources/datasets) and at the European Nucleotide Archive under accession number PRJEB81360. Genetic interactions and networks can be explored on an interactive dashboard (https://re-solute.eu/resources/dashboards/genomics).

The source data of this paper are collected in the following database record: biostudies:S-SCDT-10_1038-S44320-025-00105-5.

# Peer review information

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

## Acknowledgements

We would like to thank Jeffrey E. Barrick for providing reagents to generate self-purified Pfu-Sso7d Polymerase and William Sellers for providing the pWRS_1001 plasmid. This study received funding from the RESOLUTE consortium. RESOLUTE has received funding from the Innovative Medicines Initiative 2 Joint Undertaking under grant agreement No 777372. This Joint Undertaking receives support from the European Union's Horizon 2020 research and innovation program and EFPIA. This article reflects only the authors' views and neither IMI nor the European Union and EFPIA are responsible for any use that may be made of the information contained therein. The last year of work, including validation of data and writing of the manuscript was supported mainly by the Austrian Academy of Sciences. GS-F was supported by the Austrian Academy of Sciences throughout.

## Author contributions

**Gernot Wolf**: Conceptualization; Data curation; Formal analysis; Supervision; Validation; Investigation; Visualization; Methodology; Writing—original draft; Project administration; Writing—review and editing. **Philipp Leippe**: Data curation; Formal analysis; Validation; Investigation; Visualization; Methodology; Writing—original draft; Writing—review and editing. **Svenja Onstein**: Investigation; Methodology. **Ulrich Goldmann**: Data curation; Formal analysis; Funding acquisition. **Fabian Frommelt**: Formal analysis; Investigation; Methodology. **Shao Thing Teoh**: Investigation. **Enrico Girardi**: Conceptualization; Funding acquisition; Writing—review and editing. **Tabea Wiedmer**: Conceptualization; Supervision; Project administration; Writing—

review and editing. **Giulio Superti-Furga**: Conceptualization; Resources; Supervision; Funding acquisition; Methodology; Project administration; Writing —review and editing.

Source data underlying figure panels in this paper may have individual authorship assigned. Where available, figure panel/source data authorship is listed in the following database record: biostudies:S-SCDT-10_1038-S44320-025-00105-5.

## Disclosure and competing interests statement

GS-F is co-founder and owns shares of Solgate GmbH, an SLC-focused company. EG is the CSO of Solgate GmbH.

# Expanded View Figures

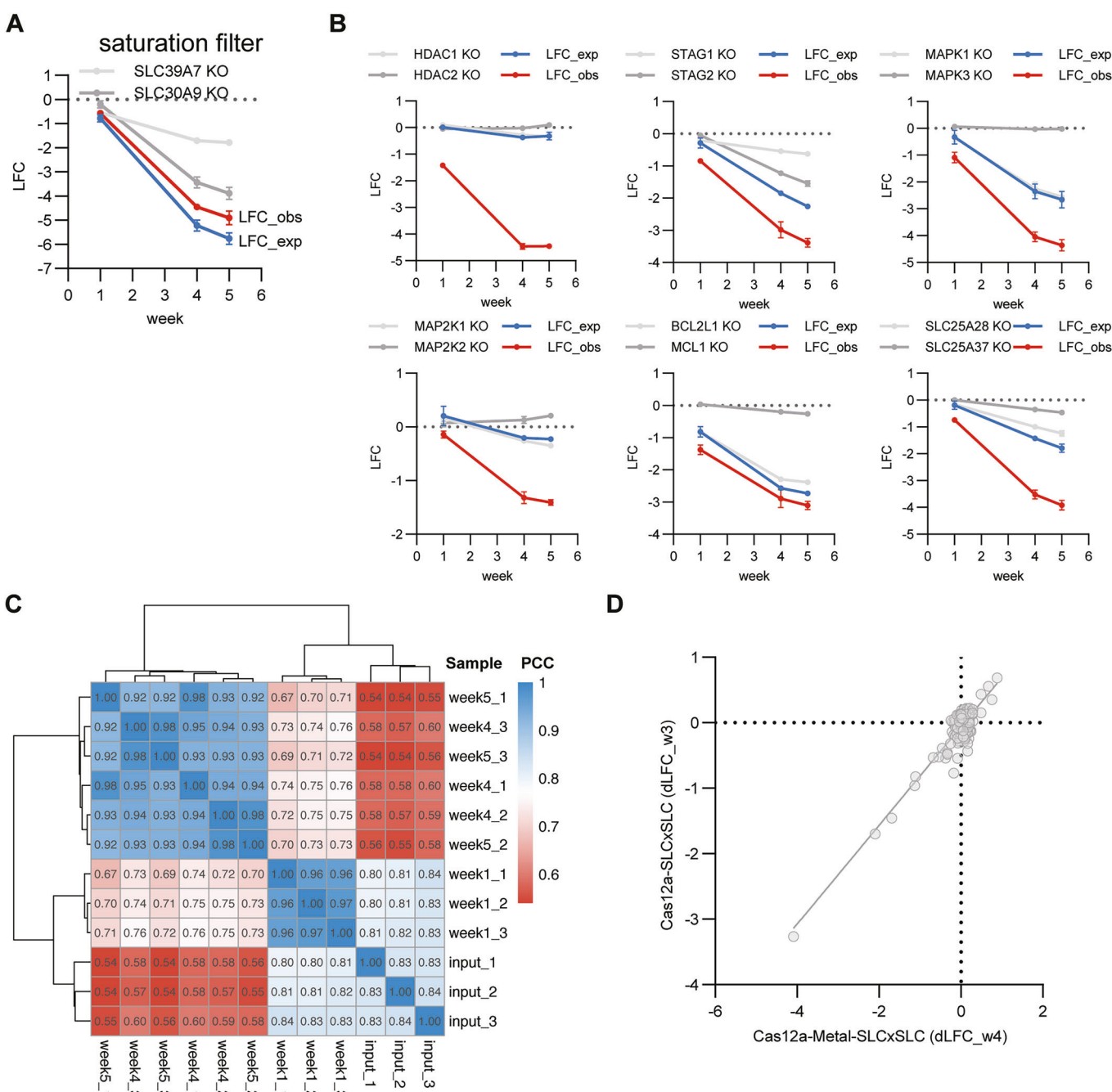

**Figure EV1.   Workflow for genetic interaction mapping of SLC transporters and benchmark screen targeting 21 metal SLC transporters plus controls (Cas12a-Metal-SLCxSLC).**

(**A**) Example of an interaction filtered out by the saturation filter, SLC39A7-SLC30A9, because LFC_obs < (SLC30A9 KO and SLC39A7). Mean ± SD of three replicates. (**B**) Change of LFC values over time for synthetic lethal positive control pairs. Mean ± SD of three replicates. (**C**) Clustered Pearson Correlation Coefficient (PCC) matrix of raw sgRNA read counts for Cas12a-Metal-SLCxSLC replicates at different time points. (**D**) dLFC values (week 4 vs week 3) for the 210 interactions (without controls) that were part of both the Cas12a-Metal-SLCxSLC benchmark screen and the SLC superfamily-wide Cas12a-SLCxSLC screen, showing highly correlated values.

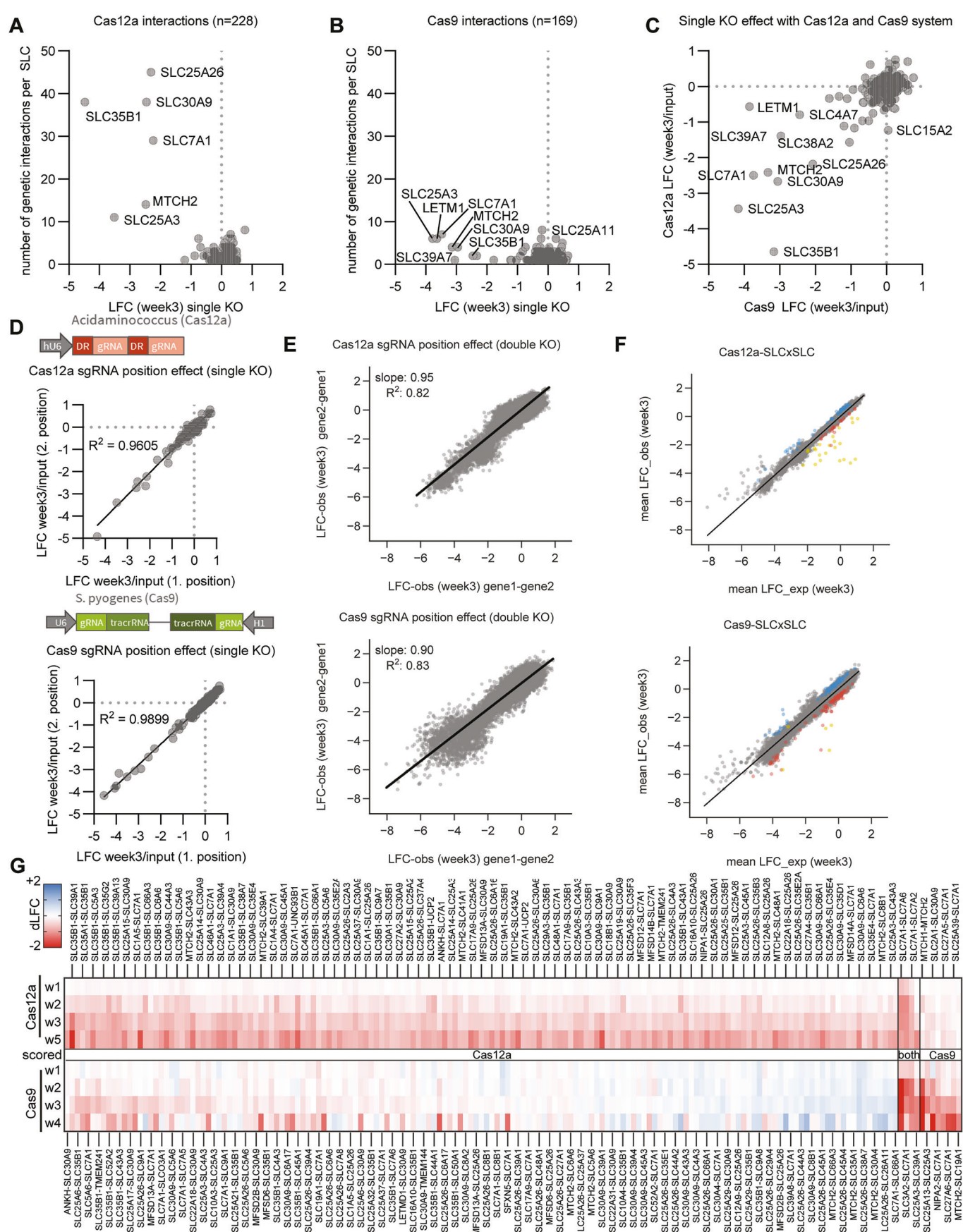

**Figure EV2. Genetic Interactions among SLC transporters in HCT 116 cells.**

(A) Number of genetic interactions per SLC in the Cas12a-SLCxSLC screen. (B) Number of genetic interactions per SLC in the Cas9-SLCxSLC screen. (C) Comparison of single KO effects between the Cas12a-SLCxSLC and Cas9-SLCxSLC screen. (D) Comparison of the effect of sgRNA orientation on single KO LFCs for the Cas12a-SLCxSLC and Cas9-SLCxSLC screen. (E) Comparison of the effect of sgRNA orientation on double KO effects for the Cas12a-SLCxSLC and Cas9-SLCxSLC screen. (F) Comparison of the observed double KO effect at week3 (LFC_obs (week3)) and the expected double KO effect from the sum of both single KO effects (LFC_exp (week3)). Yellow: control pairs, blue: scored synthetic viable interactions, red: scored synthetic lethal interactions. Interactions with the six most essential SLCs were not colored. (G) Overview of 170 interactions involving six frequently scoring SLCs that were essential as single KO in both screens based on a LFC (single KO) < −2. 158 of these interactions were identified in the Cas12a-SLCxSLC screen, while only four were found in both screens.

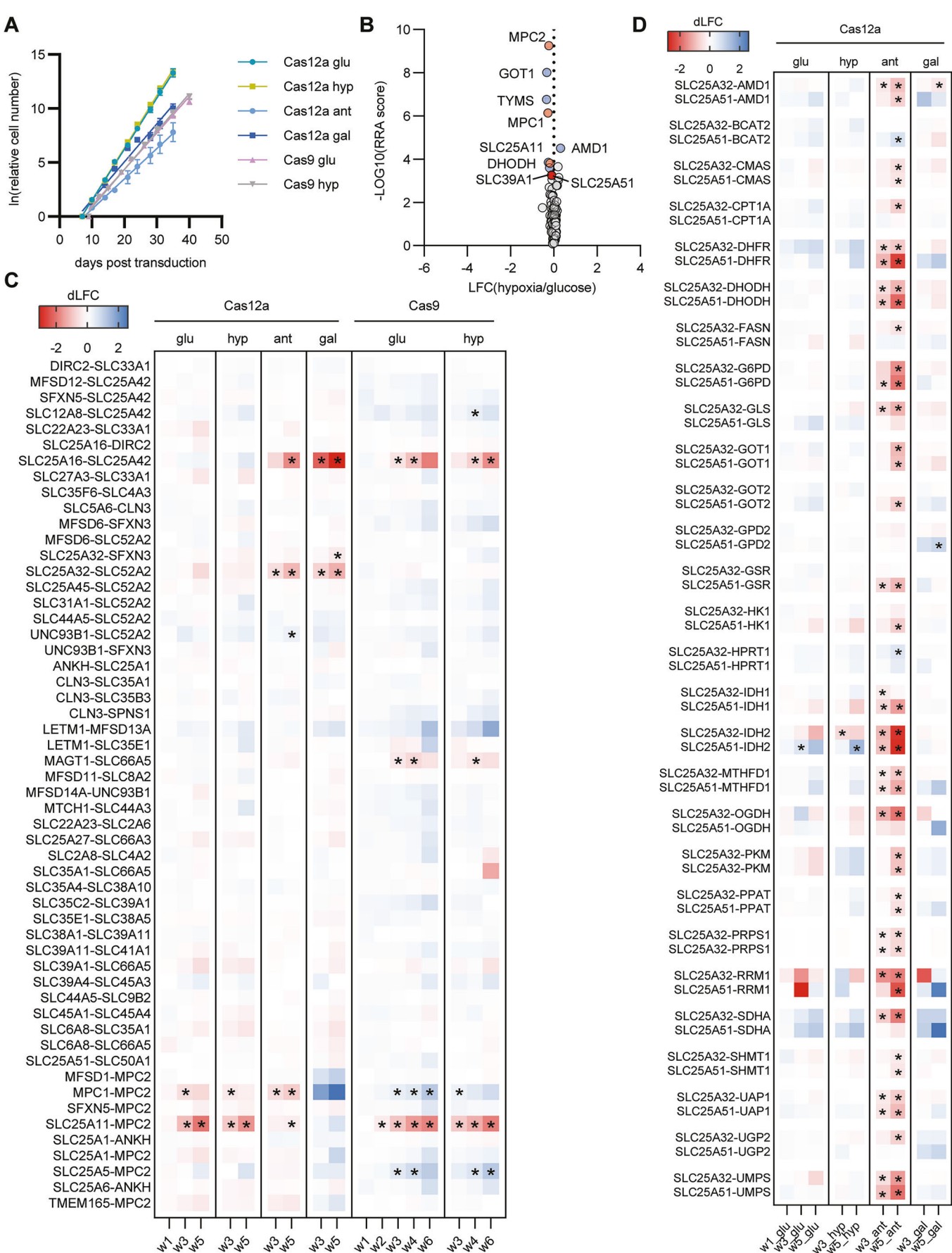

**Figure EV3.  Growth rates, single knockout effects and dynamics of genetic interactions for the SLCxEnzyme screens.**

(A) Growth rates of HCT116-Cas12a and HCT116-Cas9 cells following library transduction. Mean ± SD of three replicates. (B) Comparison of single KO effects in hypoxia versus glucose conditions. (C) Heatmap of dLFC values over time for the 54 SLC-SLC pairs additionally included in the SLCxEnzyme library. Stars indicate significance (padj < 0.1 in all three replicates). (D) Heatmap of dLFC values over time for SLC25A32 and SLC25A51 interactions. Displayed are all interactions that were significant for either SLC25A32 or SLC25A51. Stars indicate significance (padj < 0.1 in all three replicates).

                                      

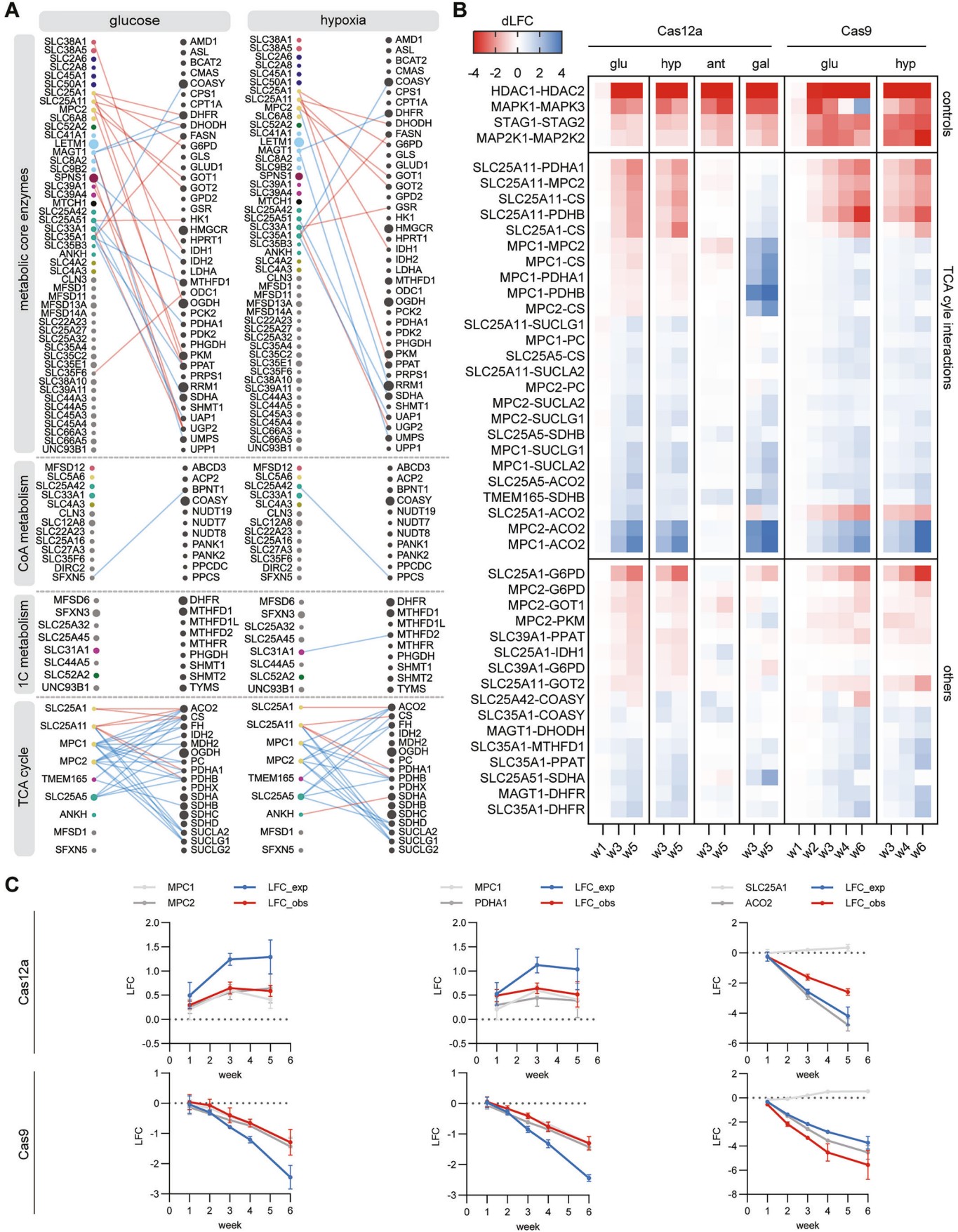

◀ **Figure EV4. Comparison of SLCxEnzyme screens using CRISPR-Cas9 and CRISPR-Cas12a systems.**

(A) Orthogonal CRISPR-Cas9 screen testing the same gene pairs as shown in Fig. 3D but only in two growth conditions. (B) Side-by-side time course dLFC values genetic interactions that were detected in either the Cas12a-SLCxEnzyme, the Cas9-SLCxEnzyme or both screens. (C) Some interactions are buffering in both Cas12a and Cas9 but in opposite directions, likely due to clonal differences between the HCT116-Cas12a and -Cas9 clones used (e.g., MPC1-MPC2 and MPC1-PDHA). This results in the same genetic interaction classification (lethal vs viable) while exhibiting opposite growth phenotypes in the Cas12a- vs the Cas9- clone. A few interactions disagreed in classification, e.g., SLC25A1-ACO2. Mean ± SD of three replicates.

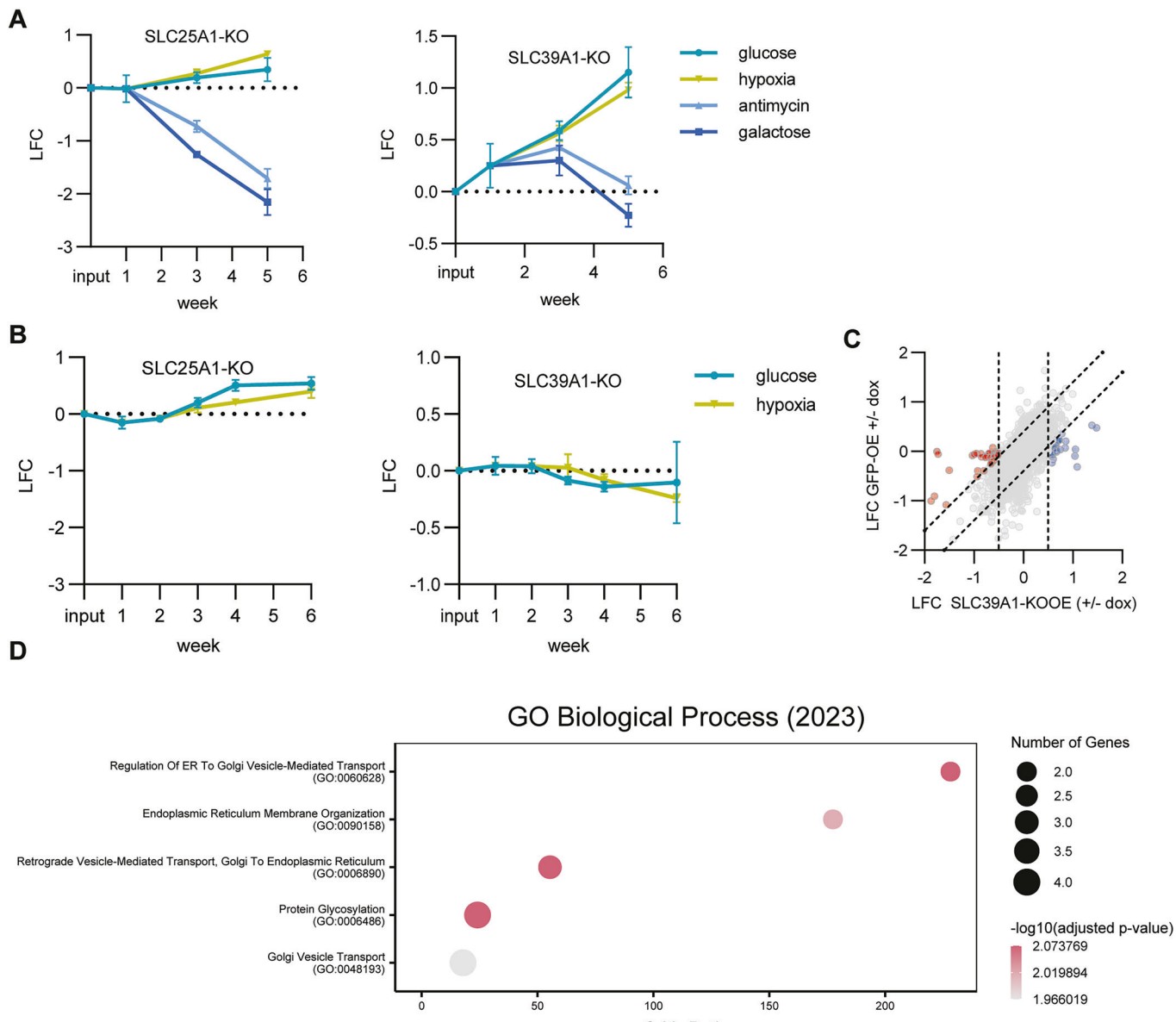

**Figure EV5. Genetic interactions and transcriptomic changes of SLC39A1 and SLC25A1.**

(A) Single KO effects of SLC25A1 and SLC39A1 across different growth conditions in the Cas12a-SLCxEnzyme screen. Mean ± SD of three replicates. (B) Single KO effects of SLC25A1 and SLC39A1 across different growth conditions in the Cas9-SLCxEnzyme screen. Mean ± SD of three replicates. (C) Transcriptomics changes in HCT116-SLC39A1-KO-OE cells after doxycycline-induced expression compared against GFP-OE control, for exclusion of effects caused by doxycycline incubation. (D) Gene ontology enrichment of the protein-protein interaction network shown in Fig. 5D. Significance was determined using Fisher's exact test and *p* values were adjusted for multiple testing using the Benjamini-Hochberg method.

