## [Peer Review File · Molecular Systems Biology]

The genetic interaction map of the human solute carrier superfamily

Gernot Wolf, Philipp Leippe, Svenja Onstein, Ulrich Goldmann, Fabian Frommelt, Shao Teoh, Enrico Girardi, Tabea Wiedmer, and Giulio Superti-Furga

Corresponding author(s): Giulio Superti-Furga (gsuperti@cemm.oeaw.ac.at)

Review Timeline:

Submission Date:	17th Oct 24
Editorial Decision:	3rd Dec 24
Revision Received:	30th Jan 25
Editorial Decision:	17th Mar 25
Revision Received:	28th Mar 25
Accepted:	31st Mar 25

Editor: Jingyi Hou

Transaction Report:

3rd Dec 2024

Manuscript Number: MSB-2024-12685

Title: The genetic interaction map of the human solute carrier superfamily

Author: Gernot Wolf

Philipp Leippe

Svenja Onstein

Ulrich Goldmann

Fabian Frommelt

Shao Teoh

Enrico Girardi

Tabea Wiedmer

Giulio Superti-Furga

Dear Guilio,

Thank you again for submitting your work to Molecular Systems Biology. I apologize for the somewhat slow process, which was due to delays in obtaining the referee reports. We have now heard back from the three reviewers who agreed to evaluate your manuscript. As you will see from the reports below, the reviewers are overall positive about the study. They raise, however, a series of concerns, which we would ask you to address in a major revision.

I think that the recommendations of the reviewers are rather clear so there is no need to repeat the points listed below. All issues raised by the reviewers need to be satisfactorily addressed. As you may already know, our editorial policy allows in principle a single round of major revision, so it is essential to provide responses to the reviewers' comments that are as complete as possible. Please feel free to contact me in case you would like to discuss in further detail any of the issues raised by the reviewers.

On a more editorial level, we would ask you to address the following issues:

- Please provide a .docx formatted version of the manuscript text (including legends for main figures, EV figures and tables). Please make sure that the changes are highlighted to be clearly visible.
- Please provide individual production quality figure files as .eps, .tif, .jpg (one file per figure).
- Please provide a .docx formatted letter INCLUDING the reviewers' reports and your detailed point-by-point responses to their comments. As part of the EMBO Press transparent editorial process, the point-by-point response is part of the Review Process File (RPF), which will be published alongside your paper.
- Please note that all corresponding authors are required to supply an ORCID ID for their name upon submission of a revised manuscript.
- We replaced Supplementary Information with Expanded View (EV) Figures and Tables that are collapsible/expandable online (see examples in <http://msb.embopress.org/content/11/6/812>). A maximum of 5 EV Figures can be typeset. EV Figures should be cited as 'Figure EV1, Figure EV2' etc... in the text and their respective legends should be included in the main text after the legends of regular figures.

Additional Tables/Datasets should be labeled and referred to as Table EV1, Dataset EV1, etc. Legends have to be provided in a separate tab in case of .xls files. Alternatively, the legend can be supplied as a separate text file (README) and zipped together with the Table/Dataset file.

For the figures and tables that you do NOT wish to display as Expanded View figures, they should be bundled together with their legends in a single PDF file called *Appendix*, which should start with a short Table of Content. Each legend should be below the corresponding Figure/Table in the Appendix. Appendix figures and tables should be referred to in the main text as: "Appendix Figure S1, Appendix Figure S2, Appendix Table S1" etc. See detailed instructions regarding expanded view here: <https://www.embopress.org/page/journal/17444292/authorguide#expandedview>.

- Before submitting your revision, primary datasets (and computer code, where appropriate) produced in this study need to be deposited in an appropriate public database (see <http://msb.embopress.org/authorguide-dataavailability> <https://www.embopress.org/page/journal/17444292/authorguide#dataavailability>).
- Please remember to provide a reviewer password if the datasets are not yet public.
- The accession numbers and database should be listed in a formal "Data Availability" section (placed after Materials & Method)

that follows the model below (see also <https://www.embopress.org/page/journal/17444292/authorguide#dataavailability>). Please note that the Data Availability Section is restricted to new primary data that are part of this study.

Data availability

-At EMBO Press we ask authors to provide source data for the main figures. Our source data coordinator will contact you to discuss which figure panels we would need source data for and will also provide you with helpful tips on how to upload and organize the files.

- Our journal encourages inclusion of *data citations in the reference list* to directly cite datasets that were re-used and obtained from public databases. Data citations in the article text are distinct from normal bibliographical citations and should directly link to the database records from which the data can be accessed. In the main text, data citations are formatted as follows: "Data ref: Smith et al, 2001". In the Reference list, data citations must be labeled with "[DATASET]". A data reference must provide the database name, accession number/identifiers and a resolvable link to the landing page from which the data can be accessed at the end of the reference. Further instructions are available at .

- We updated our journal's competing interests policy in January 2022 and request authors to consider both actual and perceived competing interests. Please review the policy <https://www.embopress.org/competing-interests> and update your competing interests if necessary.

Please use the heading "Disclosure statement and competing interests".

- All Materials and Methods need to be described in the main text using our 'Structured Methods' format. According to this format, the Methods section includes a Reagents and Tools Table (listing key reagents, experimental models, software and relevant equipment and including their sources and relevant identifiers) followed by a Methods and Protocols section describing the methods, ideally using a step-by-step protocol format. The aim is to facilitate adoption of the methodologies across labs.

Please download and fill our Reagents and Tools Table template (.docx), which you can find in our author guidelines:

<https://www.embopress.org/page/journal/17444292/authorguide#structuredmethods>.

-Regarding data quantification:

Please ensure to specify the name of the statistical test used to generate error bars and P values, the number (n) of independent experiments (please specify technical or biological replicates) underlying each data point and the test used to calculate p-values in each figure legend. Discussion of statistical methodology can be reported in the materials and methods section, but figure legends should contain a basic description of n, P and the test applied.

Graphs must include a description of the bars and the error bars (s.d., s.e.m.).

- Please provide a "standfirst text" summarizing the study in one or two sentences (approximately 250 characters, including space), three to four "bullet points" highlighting the main findings and a "synopsis image" (550px width and 400-600 px height, PNG format) to highlight the paper on our homepage.

Here are a couple of examples:

<https://www.embopress.org/doi/10.15252/msb.20199356>

<https://www.embopress.org/doi/10.15252/msb.20209475>

<https://www.embopress.org/doi/10.15252/msb.209495>

When you resubmit your manuscript, please download our CHECKLIST (<https://www.embopress.org/pb-assets/embosite/EMBO%20Press%20Author%20Checklist-1642513524327.xlsx>) and include the completed form in your submission.

Please note that the Author Checklist will be published alongside the paper as part of the transparent process (<https://www.embopress.org/page/journal/17444292/authorguide#transparentprocess>).

If you feel you can satisfactorily deal with these points and those listed by the referees, you may wish to submit a revised version of your manuscript. Please attach a covering letter giving details of the way in which you have handled each of the points raised by the referees. A revised manuscript will be once again subject to review and you probably understand that we can give you no guarantee at this stage that the eventual outcome will be favorable.

I look forward to receiving the revised manuscript soon.

Kind regards,
Jingyi

Jingyi Hou, PhD
Scientific Editor
Molecular Systems Biology

We realize that it is difficult to revise to a specific deadline. In the interest of protecting the conceptual advance provided by the work, we recommend a revision within 3 months (3rd Mar 2025). Please discuss the revision progress ahead of this time with the editor if you require more time to complete the revisions. Use the link below to submit your revision:

IMPORTANT: When you send your revision, we will require the following items:

1. the manuscript text in LaTeX, RTF or MS Word format
2. a letter with a detailed description of the changes made in response to the referees. Please specify clearly the exact places in the text (pages and paragraphs) where each change has been made in response to each specific comment given
3. three to four 'bullet points' highlighting the main findings of your study
4. a short 'blurb' text summarizing in two sentences the study (max. 250 characters)
5. a 'thumbnail image' (550px width and max 400px height, Illustrator, PowerPoint or jpeg format), which can be used as 'visual title' for the synopsis section of your paper.
6. Please include an author contributions statement after the Acknowledgements section (see <https://www.embopress.org/page/journal/17444292/authorguide>)
7. Please complete the CHECKLIST available at (<https://bit.ly/EMBOPressAuthorChecklist>). Please note that the Author Checklist will be published alongside the paper as part of the transparent process (<https://www.embopress.org/page/journal/17444292/authorguide#transparentprocess>).
8. When assembling figures, please refer to our figure preparation guideline in order to ensure proper formatting and readability in print as well as on screen:
<https://bit.ly/EMBOPressFigurePreparationGuideline>
See also figure legend guidelines: <https://www.embopress.org/page/journal/17444292/authorguide#figureformat>
9. Please note that corresponding authors are required to supply an ORCID ID for their name upon submission of a revised manuscript (EMBO Press signed a joint statement to encourage ORCID adoption). (<https://www.embopress.org/page/journal/17444292/authorguide#editorialprocess>)
Currently, our records indicate that the ORCID for your account is 0000-0002-0570-1768.

Please click the link below to modify this ORCID:
Link Not Available

11. Include a Reagents and Tools Table as part of the Methods section, which can be downloaded from our author guidelines (<https://www.embopress.org/page/journal/17444292/authorguide#structuredmethods>)

*** PLEASE NOTE *** As part of the EMBO Press transparent editorial process initiative (see our Editorial at <https://dx.doi.org/10.1038/msb.2010.72>), Molecular Systems Biology publishes online a Review Process File with each accepted manuscripts. This file will be published in conjunction with your paper and will include the anonymous referee reports, your point-by-point response and all pertinent correspondence relating to the manuscript. If you do NOT want this File to be published, please inform the editorial office at msb@embo.org within 14 days upon receipt of the present letter.

Reviewer #1:

Summary

This manuscript describes a large-scale genetic interaction analysis involving human solute carrier (SLC) superfamily, using both CRISPR-Cas9 and -Cas12a gene editing in human colon carcinoma cells (HCT116). The authors explored the effects of these interactions on cell growth and metabolism under varying conditions, including hypoxia and altered glucose levels. This study reveals a complex network of interactions among SLCs and metabolic enzymes, providing insights into the functional relationships between these proteins and their roles in cellular homeostasis.

The manuscript describes several different GI screens:

21x21 SLC gene matrix using Cas12a

- CRISPR-Cas12a targeting 210 pairs of SLC metal transporter genes expressed in HCT116
- 3 gRNAs/SLC gene, each individual guide also paired with a gRNA targeting Olfactory receptor gene as a negative control

258x258 SLC gene matrix using Cas12a

- 33,153 SLC x SLC pairs expressed in HCT116
- 228 GIs (0.69%), 183 negative and 43 positive.
- Majority of GIs (159/228) involved SLCs with single mutant fitness effects

258x258 SLC gene matrix using Cas9

- 33,153 SLC x SLC pairs expressed in HCT116
- 169 GIs (0.51%), 93 negative and 76 positive.
- Most GIs from Cas9 screen involved genes lacking single mutant effects (12/169). The authors suggest that Cas12 essential gene interactions may need to be approached with caution

SLC gene x metabolic enzyme screens. Designed 4 combinatorial libraries covering different aspects of metabolism using Cas12a and Cas9

- Library 1: 40 metabolic core enzymes x 49 SLC
- Library 2: 9 CoA metabolism genes x 8 SLCs
- Library 3: 11 one carbon metabolism enzymes x 13 SLCs
- Library 4: 17 TCA enzymes x 9 SLCs
- In total, 2340 gene pairs were tested for GIs in 4 conditions (glucose, hypoxia, antinomycin A (ETC III inhibitor), galactose)

General Comments

This is an impressive study, describing an extensive series of CRISPR-based genetic interaction screens involving SLC genes. Several different screens focussed on a defined set of genes were designed and implemented. The screens varied in their hit rate, but they collectively provide powerful insight into genetic network analysis. The combined results outline a new resource for solute carrier gene function, mapping functional connections within members of the SLC family and between metabolic pathways and SLC family genes. This study also illustrates a general model for addressing mechanistic connections within a large, functionally redundant, gene family.

Specific Comments

Screens were repeated using both Cas12a and Cas9 but the extent of overlap varied by screen. For example, the authors observe ~40% overlap among the SLC x metabolic enzyme networks but less than ~10% overlap in the SLC x SLC network. Genes with stronger single mutant fitness (SMF) defects (including essential SLC genes) showed more interactions in Cas12 screens, while GIs identified with Cas9 involved mostly nonessential genes without single mutant defects.

In both yeast and human cells, usually nonessential genes whose inactivation is associated with a SMF defect show more genetic interactions than the average gene. Functionally redundant paralogs are an unusual case. If the paralogs have not diverged functionally, they may only show one strong digenic interaction with their paralog. The transient nature of the CRISPR screens means that genetic interactions can be scored for essential genes, but it will depend on the rate at which the guide drops out of the pooled mutant library.

It might be useful to further examine the overlap of the equivalent Cas12a and Cas9 screens in the context of nonessential SLC gene x SLC gene interactions, interactions involving essential genes, interactions involving nonessential genes with a SMF defect in detail.

Reviewer #2:

Summary

The manuscript "The genetic interaction map of the human solute carrier superfamily" by Wolf and colleagues systematically investigates functional relationship of solute carrier (SLC) superfamily members via large-scale combinatorial gene perturbation in cultured human cancer cells. The authors screen different sets of SLC-SLC or SLC-Enzyme pairs using combinatorial gRNA libraries both guiding Cas12a or, alternatively, Cas9 to knock out targeted genes. Experimentally, this work was done by delivering gRNA libraries into HCT116 cells stably expressing either Cas9 or Cas12a. The work initially explores a small combinatorial perturbation set of metal-transporting SLCs to establish the experimental system. They continue by perturbing all possible pairs among the 258 expressed SLC superfamily members both using Cas9 and Cas12a. Finally, this work co-perturbs SLC genes with biologically related enzymes, this time in four distinct growth conditions that challenge and potentially rewire cellular energy metabolism. In this experiment, Cas9 combinatorial perturbations in two conditions could not be measured due to severe growth defects of the cell population. This study presents a part of an effort of generating orthogonal omics data around SLC biology. Together those data provide a unique, comprehensive and browsable data foundation for systematically exploring functional relationships within the SLC superfamily.

General remarks

The work presented by Wolf and colleagues correctly identifies the need for systematically mapping functional relations between genes and explores the SLC superfamily using combinatorial CRISPR screening. It continues a trend of mid-sized combinatorial gene perturbation screens in human cells, which aim at generating snap shots of the vast human genetic interaction network, which is likely cell type and environment-dependent and cannot be measured as a whole. In contrast to several previous studies in the field, which predominantly aimed at improving gene perturbation tools, this work utilized existing tools to record biologically useful data. The authors chose both a Cas9 and a Cas12a platform. While the cell clones used for screening seemed to display slightly different phenotypic characteristics, which somewhat limits the generality of the data, it emphasizes a most crucial point of such studies, namely the uncertainty of the model systems. In other words, I very much like the aspect that several technical limitations are communicated openly and believe that this is crucial for wide re-usability of the data. One central aspect of the current study is that genetic interactions are context specific. Conceptually, this is an important and in human cells relatively novel aspect to investigate, because genetic interaction studies in yeast suggest a robustness of such networks to environmental stimuli (DOI: 10.1126/science.abf8424). Since initial large-scale studies in a multicellular model organism, the fruit fly, has found substantial rewiring genetic interactions within a stimulated signaling network (10.1016/j.cels.2017.10.015), vast rewiring could indeed represent organization principles in cells derived from multicellular organisms. Therefore, it would be interesting to investigate if rewiring in the SLC genetic interaction network in human cells is pervasive or rather negligible. While I expect this work to become a central piece for systematic exploration of SLC biology, clarifying the conditional nature of genetic interactions of SLCs could further expand the viewership of this work, but would require a slightly more thorough statistical presentation. Overall, all key conclusions are solidly based on the data recorded in this study. I believe that this work is of very high interest to a wide audience and I would support its publication at Molecular Systems Biology after a small number of concerns have been addressed.

Major comments

1. Data reproducibility. Please visualize the reproducibility of single gene perturbation fitness effects and genetic interaction scores for ALL data sets produced. Please also see major concern 2 for more extended visualization of reproducibility. gRNA abundance reproducibility as shown in Figure EV1C can be informative but does not replace the above mentioned. Along those lines, since the Pearson correlation coefficient of 0.42 was referred to as 'moderate' on page 6, I would like to advise the author to refer to previous work (doi.org/10.1016/j.cels.2023.04.003) demonstrating the relation between hit density (e.g. GI density among screened gene pairs) and the expected range of correlation coefficients. This work would generally help place the reproducibility of this work into context and would possibly support a strong conclusion about data reproducibility despite moderate correlation coefficients.
2. AB-BA agreement. The authors report the number of possible pairs as $\{X \text{ choose } 2\}$. For instance, they report 210 unique pairs among 21 genes. However, I suspect that all gene pairs are targeted in two orientations of gRNAs on the plasmid: geneA - geneB (AB) and geneB-geneA (BA). If this indeed was the case, we had for the 210 pairs another 210 pairs where the orientation is inversed as well as 21 'self-interaction' scores (geneA-geneA, geneB-geneB, ...). While the latter can be an informative data quality metric too, I would especially emphasize that reporting the AB-BA agreement of genetic interaction scores is a crucial reproducibility metric. This will likely be different from the effect the orientation has on single KO effects reported in Figure EV2D (also since single KO effects have a different density and effect size).
3. Context similarity and differences. I would like to see a more extensive statistical assessment of genetic interactions common and private to the experimental conditions. More specifically, it would be important to learn how genetic interactions change in different conditions given the reproducibility in the same condition.

Minor comments

1. It is unclear what the mentioned Pearson correlation coefficients 'across time points' (0.76-0.91) on page 5 exactly refer to. Figure EV1D is cited in this context but only shows one data population and the comparison does not compare time points. In this regard, time points of the same replicate in a CRISPR screen often tend to be very highly correlation.
2. Please add description of error bars in Figure 1E, F and Figure EV1A, B and all alike figures in the legends.
3. Please label samples (week_x_x), units and the data represented in Figure EV1C. The legends mention gRNA abundance - is this normalized?
4. Figure EV2B. Please move dot labels so they are readable.
5. I would appreciate illustrations (e.g. in extended view figures) that show how single KO effects of each tested gene, such as

- the values on the x-axis of Figure EV2A, are scattered against the combinatorial effects of those genes with different query genes (I understand that the experimental gene matrix is symmetric and all genes can be considered a query gene too).
6. Please expand the description in the methods of how genetic interactions were scored including how many data points go into which statistical test.
 7. Please provide the code used for generating the genetic interaction scores.

Reviewer #3:

In this comprehensive study, the author employed a dual-knockout strategy to illuminate the intricate interplay between solute carrier proteins (SLCs) and specific metabolic enzymes. Their findings reveal an extensive network of connections among various SLCs and their corresponding metabolic enzymes, suggesting a complex and adaptive cellular metabolism capable of re-routing pathways following the loss of a single gene. However, under conditions where dual knockouts target genes with closely related functions, cell survival may be severely compromised, highlighting the significance of redundant metabolic routes. Their analysis uncovers numerous intriguing associations between SLCs and metabolic enzymes, several of which warrant further exploration. They delve into the most substantial relationships and propose mechanisms underlying these interactions, although additional research is required to validate their hypotheses.

Regarding future experiments, I offer some queries and suggestions. In the ongoing discussions, since it was noted that protein localization can vary across different systems, and considering that HCT116 was predominantly utilized as the baseline system in previous experiments, using HCT116 rather than 293T in protein localization and pull-down assay would likely bolster the credibility of the results. Given that the study posits regulation through the NF- κ B pathway, augmenting transcriptomics data with Western Blot verification of critical protein expressions would substantially enhance the article's comprehensiveness.

Moreover, refining the writing style to optimize readability is highly recommended. Clarity and precision in conveying scientific concepts, along with smooth transitions between ideas, will make the text more engaging for readers.

We advocate for the publication of this work given its substantial contribution to the field, offering a novel approach to uncovering the elusive substrates of SLCs by using these relationships, and give us a new aspect to design experimental exploration direction.

Point-by-point response for the manuscript: The genetic interaction map of the human solute carrier superfamily.

Response to reviewers of “The genetic interaction map of the human solute carrier superfamily”

Reviewer #1:

Summary

This manuscript describes a large-scale genetic interaction analysis involving human solute carrier (SLC) superfamily, using both CRISPR-Cas9 and -Cas12a gene editing in human colon carcinoma cells (HCT116). The authors explored the effects of these interactions on cell growth and metabolism under varying conditions, including hypoxia and altered glucose levels. This study reveals a complex network of interactions among SLCs and metabolic enzymes, providing insights into the functional relationships between these proteins and their roles in cellular homeostasis.

The manuscript describes several different GI screens:

21x21 SLC gene matrix using Cas12a

- CRISPR-Cas12a targeting 210 pairs of SLC metal transporter genes expressed in HCT116
- 3 gRNAs/SLC gene, each individual guide also paired with a gRNA targeting Olfactory receptor gene as a negative control

258x258 SLC gene matrix using Cas12a

- 33,153 SLC x SLC pairs expressed in HCT116
- 228 GIs (0.69%), 183 negative and 43 positive.
- Majority of GIs (159/228) involved SLCs with single mutant fitness effects

258x258 SLC gene matrix using Cas9

- 33,153 SLC x SLC pairs expressed in HCT116
- 169 GIs (0.51%), 93 negative and 76 positive.
- Most GIs from Cas9 screen involved genes lacking single mutant effects (12/169). The authors suggest that Cas12 essential gene interactions may need to be approached with caution

SLC gene x metabolic enzyme screens. Designed 4 combinatorial libraries covering different aspects of metabolism using Cas12a and Cas9

- Library 1: 40 metabolic core enzymes x 49 SLC
- Library 2: 9 CoA metabolism genes x 8 SLCs
- Library 3: 11 one carbon metabolism enzymes x 13 SLCs
- Library 4: 17 TCA enzymes x 9 SLCs
- In total, 2340 gene pairs were tested for GIs in 4 conditions (glucose, hypoxia, antinimycin A (ETC III inhibitor), galactose)

General Comments

This is an impressive study, describing an extensive series of CRISPR-based genetic interaction screens involving SLC genes. Several different screens focussed on a defined set of genes were designed and implemented. The screens varied in their hit rate, but they collectively provide powerful insight into genetic network analysis. The combined results outline a new resource for solute carrier gene function,

mapping functional connections within members of the SLC family and between metabolic pathways and SLC family genes. This study also illustrates a general model for addressing mechanistic connections within a large, functionally redundant, gene family.

We thank the reviewer for their positive summary of our work. We are pleased that the scope and depth of our CRISPR-based genetic interaction screens, as well as their potential to provide insights into SLC gene functions and their connections to metabolism, are recognized. Please find our point-by-point response below:

Specific Comments

Screens were repeated using both Cas12a and Cas9 but the extent of overlap varied by screen. For example, the authors observe ~40% overlap among the SLC x metabolic enzyme networks but less than ~10% overlap in the SLC x SLC network. Genes with stronger single mutant fitness (SMF) defects (including essential SLC genes) showed more interactions in Cas12 screens, while GIs identified with Cas9 involved mostly nonessential genes without single mutant defects.

We thank the reviewer for this interesting observation, which aligns with our own analysis. The difference in overlap may be attributed to the intentional selection of SLCxEnzyme gene pairs, which were chosen to cover specific metabolic pathways and resulted in an overrepresentation of mitochondrial carriers in the SLCxEnzyme library. These consistently exhibited a higher-than-average number of genetic interactions in both the SLCxSLC and SLCxEnzyme screens, and their interactions are arguably less affected by biological differences between HCT116-Cas9 and -Cas12a clones than other transporters (see pages 8 and 14). However, the smaller size of the SLCxEnzyme libraries compared to the SLCxSLC screens makes direct comparisons of overlap fractions inherently more challenging. However, an important conclusion emerges from this analysis: pairing SLCs with metabolic enzymes may be a more effective strategy for studying SLC biology. These interactions were not only more frequent but also more informative, as metabolic enzymes are typically better annotated, facilitating functional inferences from genetic interaction networks.

In both yeast and human cells, usually nonessential genes whose inactivation is associated with a SMF defect show more genetic interactions than the average gene. Functionally redundant paralogs are an unusual case. If the paralogs have not diverged functionally, they may only show one strong digenic interaction with their paralog. The transient nature of the CRISPR screens means that genetic interactions can be scored for essential genes, but it will depend on the rate at which the guide drops out of the pooled mutant library.

The rate of guide dropout may explain why interactions with the six most essential SLCs—defined as $LFC(\text{single KO}) < -2$ in both screens—were detected more frequently in the Cas12a screen (Figure EV2A) than in the Cas9 screen (Figure EV2B). This difference could be attributed to the higher magnitude of single KO effects observed in the Cas9 screen compared to Cas12a. Among these six genes, one (SLC25A25) exhibited similar single KO effects in both screens, while four (SLC7A1, SLC25A3, MTCH2, SLC30A9) showed stronger single KO effects in the Cas9 screen (Figure EV2C). To address this point, we have added a clarifying sentence on page 6.

It might be useful to further examine the overlap of the equivalent Cas12a and Cas9 screens in the context of nonessential SLC gene x SLC gene interactions, interactions involving essential genes, interactions involving nonessential genes with a SMF defect in detail.

We thank the reviewer for the suggestion. The majority of single SLC KOs exhibited only mild growth phenotypes (Figure EV2C), especially compared to single KOs of metabolic enzymes (Figure 3C,D; Figure EV4A). In Figure 2E, we present the outer merge of genetic interactions, excluding the six essential genes (SLC25A26, SLC7A1, MTCH2, SLC30A9, SLC25A3, and SLC35B1; LFC < -2 in both screens), showing that 14 out of 208 interactions (6.7%) were detected in both screens. Interactions involving these six essential genes are shown separately in an outer merge (Figure EV2G), where 4 out of 170 interactions (2.3%) overlapped. As discussed on pages 7 and 14, we believe these differences in genetic interaction profiles are likely to be largely attributed to clonal variations. When designing the study, the HCT116 Cas12a and Cas9 cell lines were clonally selected for high double-editing efficiency (Methods, page 32), which likely resulted in single clones with distinct genetic and metabolic background. This is reflected in the different phenotypes observed, for example, in terms of growth rates (Figure EV3A).

Reviewer #2:

Summary

The manuscript "The genetic interaction map of the human solute carrier superfamily" by Wolf and colleagues systematically investigates functional relationship of solute carrier (SLC) superfamily members via large-scale combinatorial gene perturbation in cultured human cancer cells. The authors screen different sets of SLC-SLC or SLC-Enzyme pairs using combinatorial gRNA libraries both guiding Cas12a or, alternatively, Cas9 to knock out targeted genes. Experimentally, this work was done by delivering gRNA libraries into HCT116 cells stably expressing either Cas9 or Cas12a. The work initially explores a small combinatorial perturbation set of metal-transporting SLCs to establish the experimental system. They continue by perturbing all possible pairs among the 258 expressed SLC superfamily members both using Cas9 and Cas12a. Finally, this work co-perturbs SLC genes with biologically related enzymes, this time in four distinct growth conditions that challenge and potentially rewire cellular energy metabolism. In this experiment, Cas9 combinatorial perturbations in two conditions could not be measured due to severe growth defects of the cell population. This study presents a part of an effort of generating orthogonal omics data around SLC biology. Together those data provide a unique, comprehensive and browsable data foundation for systematically exploring functional relationships within the SLC superfamily.

General remarks

The work presented by Wolf and colleagues correctly identifies the need for systematically mapping functional relations between genes and explores the SLC superfamily using combinatorial CRISPR screening. It continues a trend of mid-sized combinatorial gene perturbation screens in human cells, which aim at generating snap shots of the vast human genetic interaction network, which is likely cell type and environment-dependent and cannot be measured as a whole. In contrast to several previous studies in the field, which predominantly aimed at improving gene perturbation tools, this work utilized existing tools to record biologically useful data. The authors chose both a Cas9 and a Cas12a platform.

While the cell clones used for screening seemed to display slightly different phenotypic characteristics, which somewhat limits the generality of the data, it emphasizes a most crucial point of such studies, namely the uncertainty of the model systems. In other words, I very much like the aspect that several technical limitations are communicated openly and believe that this is crucial for wide re-usability of the data. One central aspect of the current study is that genetic interactions are context specific. Conceptually, this is an important and in human cells relatively novel aspect to investigate, because genetic interaction studies in yeast suggest a robustness of such networks to environmental stimuli (DOI: 10.1126/science.abf8424). Since initial large-scale studies in a multicellular model organism, the fruit fly, has found substantial rewiring genetic interactions within a stimulated signaling network (10.1016/j.cels.2017.10.015), vast rewiring could indeed represent organization principles in cells derived from multicellular organisms. Therefore, it would be interesting to investigate if rewiring in the SLC genetic interaction network in human cells is pervasive or rather negligible. While I expect this work to become a central piece for systematic exploration of SLC biology, clarifying the conditional nature of genetic interactions of SLCs could further expand the viewership of this work, but would require a slightly more thorough statistical presentation. Overall, all key conclusions are solidly based on the data recorded in this study. I believe that this work is of very high interest to a wide audience and I would support its publication at Molecular Systems Biology after a small number of concerns have been addressed.

We would like to thank the expert reviewer for taking the time to read our manuscript in detail, for their generally favorable assessment, and for their helpful specific comments. Please find our point-by-point response below:

Major comments

1. Data reproducibility. Please visualize the reproducibility of single gene perturbation fitness effects and genetic interaction scores for ALL data sets produced. Please also see major concern 2 for more extended visualization of reproducibility. gRNA abundance reproducibility as shown in Figure EV1C can be informative but does not replace the above mentioned. Along those lines, since the Pearson correlation coefficient of 0.42 was referred to as 'moderate' on page 6, I would like to advise the author to refer to previous work (doi.org/10.1016/j.cels.2023.04.003) demonstrating the relation between hit density (e.g. GI density among screened gene pairs) and the expected range of correlation coefficients. This work would generally help place the reproducibility of this work into context and would possibly support a strong conclusion about data reproducibility despite moderate correlation coefficients.

We appreciate the reviewer's helpful feedback and have addressed the request for additional visualization of data reproducibility by including a new "Figure Appendix S2," referenced in the Materials and Methods under "Data analysis for...". This figure presents Pearson Correlation Coefficients (PCCs) for all performed screens across four datasets: (i) sgRNA read counts of the input library, (ii) sgRNA read counts at week 3, (iii) expected double knockout log₂ fold-change values (LFC_{exp}, derived from single knockout effects), and (iv) observed double knockout effects (LFC_{obs}). The selected time points align with those shown in the main figure volcano plots to ensure consistency. We also cited the provided

reference (doi.org/10.1016/j.cels.2023.04.003) to contextualize the PCC of 0.42 previously described as "moderate." We have removed this imprecise wording, as the PCC represents a correlation of dLFC values from different biological contexts (HCT 116-Cas9 and HCT 116-Cas12a cells) and falls within a normal range, as highlighted in the referenced work.

2. AB-BA agreement. The authors report the number of possible pairs as $\{X \text{ choose } 2\}$. For instance, they report 210 unique pairs among 21 genes. However, I suspect that all gene pairs are targeted in two orientations of gRNAs on the plasmid: geneA -geneB (AB) and geneB-geneA (BA). If this indeed was the case, we had for the 210 pairs another 210 pairs where the orientation is inverted as well as 21 'self-interaction' scores (geneA-geneA, geneB-geneB, ...). While the latter can be an informative data quality metric too, I would especially emphasize that reporting the AB-BA agreement of genetic interaction scores is a crucial reproducibility metric. This will likely be different from the effect the orientation has on single KO effects reported in Figure EV2D (also since single KO effects have a different density and effect size).

The reviewer raises an important point about the AB-BA orientation in our screens. Indeed, the SLCxSLC screens are symmetric, with both orientations of gene pairs (AB and BA) present on the plasmid. However, the SLCxEnzyme screens are directionally paired, with SLCs as gene1 and Enzymes as gene2, making the AB-BA analysis applicable only to the SLCxSLC screens.

To address this point, we have added an AB-BA agreement analysis for both the Cas12a-SLCxSLC and Cas9-SLCxSLC screens as Fig. EV2E. In this analysis, we plotted the observed log fold changes (LFC_obs) of gene2-gene1 against gene1-gene2, using gene-level averages of the corresponding gRNA orientations. We chose to plot LFC_obs rather than genetic interaction scores (dLFC) because interactions are sparse, and plotting dLFC values would result in a difficult to interpret distribution of points around 0. Linear regression analysis on LFC_obs values shows good agreement between orientations, with the Cas12a system showing a slope of 0.95 and R^2 of 0.82, and the Cas9 system showing a slope of 0.90 and R^2 of 0.83 (Fig. EV2E). These results indicate that while the orientations agree well overall, double knockouts exhibit a slightly stronger guide position effect compared to single knockouts, despite the single knockout values being derived from guide pairings with cutting guides targeting olfactory receptors.

3. Context similarity and differences. I would like to see a more extensive statistically assessment of genetic interactions common and private to the experimental conditions. More specifically, it would be important to learn how genetic interactions change in different conditions given the reproducibility in the same condition.

We thank the reviewer for an excellent question, as it touches on a central biological question: how genetic interactions respond to changes in the environment. While direct comparison between Cas9-SLCxSLC and Cas12a-SLCxSLC screens is limited by potential Cas system-specific effects, we can analyze interaction changes across glucose, hypoxia, galactose, and antimycin conditions in the Cas12a-SLCxEnzyme screen.

Of 134 single KO effects, 41 (31%) showed substantial changes (LFC > 1) in antimycin/glucose and/or galactose/glucose comparisons (Figure 3C). This high proportion likely reflects the enrichment of

mitochondrial genes in our SLC×Enzyme library and the metabolic perturbations induced by galactose and antimycin, as we now note on page 8. Among 2,357 tested gene pairs, 186 (8%) showed significant interaction changes between conditions, with 0 changes in hypoxia, 47 in galactose, and 157 in antimycin.

This analysis combined week 3 and week 5 replicates (6 dLFC values per gene pair) for each condition, using the two-stage linear step-up procedure of Benjamini, Krieger, and Yekutieli for P value adjustment (thresholds: $P_{adj} < 0.1$, $|dLFC| > 0.3$). We visualize all condition-specific interaction changes in Figure Appendix S1, now referenced on page 9.

Minor comments

1. It is unclear what the mentioned Pearson correlation coefficients 'across time points' (0.76-0.91) on page 5 exactly refer to. Figure EV1D is cited in this context but only shows one data population and the comparison does not compare time points. In this regard, time points of the same replicate in a CRISPR screen often tend to be very highly correlation.

We thank the reviewer for highlighting this ambiguity. The intent of this sentence was to highlight that the dLFC values from the smaller Cas12a-Metal-SLC×SLC screen were well recovered in the larger Cas12a-SLC×SLC screen. The Pearson correlation coefficients refer to comparisons between dLFC values from corresponding time points in the Cas12a-Metal-SLC×SLC and larger Cas12a-SLC×SLC screens (e.g., input vs. input, week 1 vs. week 1, ...). Note that the final time point was changed from week 4 in the Metal screen to week 3 in the larger screen. Thus, in Fig. EV1D, we compare Cas12a-Metal-SLC×SLC (week 4) with Cas12a-SLC×SLC (week 3). We have revised the text and figure caption to remove ambiguity and only refer to the specific correlation that is shown in Fig EV1D.

2. Please add description of error bars in Figure 1E, F and Figure EV1A, B and all alike figures in the legends.

All error bars represent SD. This information was added to the relevant figure captions.

3. Please label samples (week_x_x), units and the data represented in Figure EV1C. The legends mention gRNA abundance - is this normalized?

We added the labels to Figure EV1C. The figure caption now clarifies that the analysis was performed on raw read counts.

4. Figure EV2B. Please move dot labels so they are readable.

We moved the dot labels to ensure readability.

5. I would appreciate illustrations (e.g. in extended view figures) that show how single KO effects of each tested gene, such as the values on the x-axis of Figure EV2A, are scattered against the combinatorial effects of those genes with different query genes (I understand that the experimental gene matrix is symmetric and all genes can be considered a query gene too).

We thank the reviewer for this suggestion. While plotting single knockout effects directly against their combinatorial effects is feasible, this would show each interaction (AB) twice - once for gene A and once for gene B. To avoid this redundancy and potential misinterpretation from plotting a single experimental value (AB) twice, we have instead added Figure EV2F, which directly compares observed double knockout effects (AB) at week 3 (LFC_obs (week 3)) with expected effects calculated as sum of single knockout effects (A+B) (LFC_exp (week 3)). As such, each gene pair is plotted once, showing how combinatorial perturbations deviate from additive expectations based on single knockout measurements, and where interactions are scored in relation to single knockout magnitude.

6. Please expand the description in the methods of how genetic interactions were scored including how many data points go into which statistical test.

We have revised the data analysis section (page 36) to provide a more detailed description of how genetic interactions were scored. Additionally, Table EV8 has been updated to clarify the number of data points used in statistical testing. For example, in the Cas12a-SLC×Enzyme screen, SLC-A_guide_1 was paired with Enzyme-B_guide_1, and SLC-A_guide_2 was paired with Enzyme-B_guide_2, with 6 sgRNAs per gene resulting in six guide combinations per gene pair. Across three replicates, this yielded 18 read counts for gene pairs, from which 18 log₂ fold changes (LFC_obs) values were calculated. These 18 LFC_obs were tested for significance against 18 LFC_exp values derived from the pairings of SLC-A and Enz-B sgRNAs with olfactory sgRNAs (i.e. single KO effects). These numbers are now explicitly listed for each screen in Table EV8.

7. Please provide the code used for generating the genetic interaction scores.

We did not use code in the traditional sense for generating genetic interaction scores. Read processing was performed through the Galaxy platform (<https://usegalaxy.org/>) using algorithms as described in the methods section. Raw read counts were subsequently processed in Microsoft Excel to calculate the log₂ fold changes for each sgRNA pair as well as the mean single KO effects (calculated for each sgRNA pair using the corresponding control sgRNA pairs). We then performed the significance testing as described in the Method section and in Table EV8 using the student's t-test function in Microsoft Excel and performed p value adjustment with GraphPad Prism. We expanded the methods section to clarify how the interaction cores were generated.

Reviewer #3:

In this comprehensive study, the author employed a dual-knockout strategy to illuminate the intricate interplay between solute carrier proteins (SLCs) and specific metabolic enzymes. Their findings reveal an extensive network of connections among various SLCs and their corresponding metabolic enzymes, suggesting a complex and adaptive cellular metabolism capable of re-routing pathways following the loss of a single gene. However, under conditions where dual knockouts target genes with closely related functions, cell survival may be severely compromised, highlighting the significance of redundant metabolic routes.

Their analysis uncovers numerous intriguing associations between SLCs and metabolic enzymes, several of which warrant further exploration. They delve into the most substantial relationships and propose

mechanisms underlying these interactions, although additional research is required to validate their hypotheses.

Regarding future experiments, I offer some queries and suggestions. In the ongoing discussions, since it was noted that protein localization can vary across different systems, and considering that HCT116 was predominantly utilized as the baseline system in previous experiments, using HCT116 rather than 293T in protein localization and pull-down assay would likely bolster the credibility of the results. Given that the study posits regulation through the NF- κ B pathway, augmenting transcriptomics data with Western Blot verification of critical protein expressions would substantially enhance the article's comprehensiveness.

Moreover, refining the writing style to optimize readability is highly recommended. Clarity and precision in conveying scientific concepts, along with smooth transitions between ideas, will make the text more engaging for readers.

We advocate for the publication of this work given its substantial contribution to the field, offering a novel approach to uncovering the elusive substrates of SLCs by using these relationships, and give us a new aspect to design experimental exploration direction.

We thank the reviewer for their thoughtful and positive feedback and are pleased that our work is recognized as a substantial contribution to the field. As highlighted, our study provides a valuable starting point for future investigations into SLC biology, including exploring the connection between SLC39A1 and mitochondrial metabolism. We agree with the reviewer that addressing this will require additional cell models and further efforts in future work. While we are actively pursuing this line of research, SLC39A1 being a metal transporter and its intracellular localization present significant experimental challenges.

17th Mar 2025

Manuscript Number: MSB-2024-12685R

Title: The genetic interaction map of the human solute carrier superfamily

Author: Gernot Wolf

Philipp Leippe

Svenja Onstein

Ulrich Goldmann

Fabian Frommelt

Shao Teoh

Enrico Girardi

Tabea Wiedmer

Giulio Superti-Furga

Dear Giulio,

Thank you for submitting your revised manuscript to Molecular Systems Biology. We have now received the enclosed report from two Reviewers who agreed to re-assess your work. As you will see below, both reviewers are overall satisfied with the revisions. Before we can accept the manuscript for publication, please address the remaining concerns from Reviewer #2 regarding the original major comment #2 and minor comment #7.

On a more editorial level, please address the following issues:

1. Please note that the figures should be removed from manuscript file. The legends should remain in the manuscript file, located below the References.
2. Funding information: all the information from the Comments box need to be included in the list of funders using the "More Funders" option.
3. Remove the "Author contributions" section from the manuscript file.
4. Please add callouts for Appendix Figure S1-S2; callout "EV Fig 3B" should be corrected to "Fig EV3B".
5. EV tables and datasets: Source file names, titles, legends and manuscript callouts all need to be updated to Dataset EV1-EV7 instead of Table EV1-EV7. The legend for Table EV8 should not be uploaded as a separate tab/sheet, but placed above the table in the Excel file. Since this is now the only EV table, it should be renamed to Table EV1 with the corresponding callouts.
6. "Declaration of interests" should be renamed to "DISCLOSURE AND COMPETING INTERESTS STATEMENT".
7. "Supplementary information" should be removed from the manuscript file.
8. Data availability: please remove the NOTE.
9. Appendix file needs to be provided in pdf format.
10. The synopsis image is too large. Please ensure it does not exceed the specified dimensions (550px width and 400-600 px height, PNG format).
11. Material and Methods" should be renamed to "Methods".
12. Section order should be corrected: Title page - Abstract & Keywords - Introduction - Results - Discussion - Methods - Data Availability - Acknowledgements - Disclosure and Competing Interests Statement - References - Figure Legends - Table(s) - Expanded View Figure Legends.
13. Please address the following comments related to figure legends:
 - Please indicate the statistical test used for data analysis in the legends of figures 1D, 2B, D,5A, EV5 D.
 - Please note that information related to n is missing in the legends of figures 1D, E, F; 5A, EV1 A, B; EV3 A, EV4 C, EV5 A,B.
 - Please note that the measure of center for the error bars needs to be defined in the legends of figures 1E, F; EV1 A, B; EV3 A, EV4 C, EV5 A, B.
 - Please note that the scale bar needs to be defined for figure 5D

Please resubmit your revised manuscript online, with a covering letter listing amendments and responses to each point raised by the referees. Please resubmit the paper ****within one month**** and ideally as soon as possible.

When you resubmit your manuscript, please download our CHECKLIST (<https://bit.ly/EMBOPressAuthorChecklist>) and include the completed form in your submission. ***Please note*** that the Author Checklist will be published alongside the paper as part of the transparent process (<https://www.embopress.org/page/journal/17444292/authorguide#transparentprocess>)

Click on the link below to submit your revised paper.

Sincerely,
Jingyi

Jingyi Hou, PhD
Senior Editor
Molecular Systems Biology

***** PLEASE NOTE ***** As part of the EMBO Press transparent editorial process initiative (see our Editorial at <https://dx.doi.org/10.1038/msb.2010.72> , Molecular Systems Biology will publish online a Review Process File to accompany accepted manuscripts. When preparing your letter of response, please be aware that in the event of acceptance, your cover letter/point-by-point document will be included as part of this File, which will be available to the scientific community. More information about this initiative is available in our Instructions to Authors. If you have any questions about this initiative, please contact the editorial office (msb@embo.org).

Reviewer #1:

The authors have addressed all my concerns.

Reviewer #2:

R2 response: Overall, the authors have addressed my concerns. As you will see below, I recommend to pay some attention to two concerns (see responses as 'R2 response'). I believe the manuscript is now suitable for publication.

Summary

The manuscript "The genetic interaction map of the human solute carrier superfamily" by Wolf and colleagues systematically investigates functional relationship of solute carrier (SLC) superfamily members via large-scale combinatorial gene perturbation in cultured human cancer cells. The authors screen different sets of SLC-SLC or SLC-Enzyme pairs using combinatorial gRNA libraries both guiding Cas12a or, alternatively, Cas9 to knock out targeted genes. Experimentally, this work was done by delivering gRNA libraries into HCT116 cells stably expressing either Cas9 or Cas12a. The work initially explores a small combinatorial perturbation set of metal-transporting SLCs to establish the experimental system. They continue by perturbing all possible pairs among the 258 expressed SLC superfamily members both using Cas9 and Cas12a. Finally, this work co-perturbs SLC genes with biologically related enzymes, this time in four distinct growth conditions that challenge and potentially rewire cellular energy metabolism. In this experiment, Cas9 combinatorial perturbations in two conditions could not be measured due to severe growth defects of the cell population. This study presents a part of an effort of generating orthogonal omics data around SLC biology. Together those data provide a unique, comprehensive and browsable data foundation for systematically exploring functional relationships within the SLC superfamily.

General remarks

The work presented by Wolf and colleagues correctly identifies the need for systematically mapping functional relations between genes and explores the SLC superfamily using combinatorial CRISPR

screening. It continues a trend of mid-sized combinatorial gene perturbation screens in human cells, which aim at generating snap shots of the vast human genetic interaction network, which is likely cell type and environment-dependent and cannot be measured as a whole. In contrast to several previous studies in the field, which predominantly aimed at improving gene perturbation tools, this work utilized existing tools to record biologically useful data. The authors chose both a Cas9 and a Cas12a platform. While the cell clones used for screening seemed to display slightly different phenotypic characteristics, which somewhat limits the generality of the data, it emphasizes a most crucial point of such studies, namely the uncertainty of the model systems. In other words, I very much like the aspect that several technical limitations are communicated openly and believe that this is crucial for wide re-usability of the data. One central aspect of the current study is that genetic interactions are context specific. Conceptually, this is an important and in human cells relatively novel aspect to investigate, because genetic interaction studies in yeast suggest a robustness of such networks to environmental stimuli (DOI: 10.1126/science.abf8424). Since initial large-scale studies in a multicellular model organism, the fruit fly, has found substantial rewiring genetic interactions within a stimulated signaling network (10.1016/j.cels.2017.10.015), vast rewiring could indeed represent organization principles in cells derived from multicellular organisms. Therefore, it would be interesting to investigate if rewiring in the SLC genetic interaction network in human cells is pervasive or rather negligible. While I expect this work to become a central piece for systematic exploration of SLC biology, clarifying the conditional nature of genetic interactions of SLCs could further expand the viewership of this work, but would require a slightly more thorough statistical presentation. Overall, all key conclusions are solidly based on the data recorded in this study. I believe that this work is of very high interest to a wide audience and I would support its publication at Molecular Systems Biology after a small number of concerns have been addressed.

Authors: We would like to thank the expert reviewer for taking the time to read our manuscript in detail, for their generally favorable assessment, and for their helpful specific comments. Please find our point-by-point response below:

Major comments

1. Data reproducibility. Please visualize the reproducibility of single gene perturbation fitness effects and genetic interaction scores for ALL data sets produced. Please also see major concern 2 for more extended visualization of reproducibility. gRNA abundance reproducibility as shown in Figure EV1C can be informative but does not replace the above mentioned. Along those lines, since the Pearson correlation coefficient of 0.42 was referred to as 'moderate' on page 6, I would like to advice the author to refer to previous work (doi.org/10.1016/j.cels.2023.04.003) demonstrating the relation between hit density (e.g. GI density among screened gene pairs) and the expected range of correlation coefficients. This work would generally help place the reproducibility of this work into context and would possibly support a strong conclusion about data reproducibility despite moderate correlation coefficients.

Authors: We appreciate the reviewer's helpful feedback and have addressed the request for additional visualization of data reproducibility by including a new "Figure Appendix S2," referenced in the Materials and Methods under "Data analysis for...". This figure presents Pearson Correlation Coefficients (PCCs) for all performed screens across four datasets: (i) sgRNA read counts of the input library, (ii) sgRNA read counts at week 3, (iii) expected double knockout log₂ fold-change values (LFC_{exp}, derived from single knockout effects), and (iv) observed double knockout effects (LFC_{obs}). The selected time points align with those shown in the main figure volcano plots to ensure consistency. We also cited the provided reference (doi.org/10.1016/j.cels.2023.04.003) to contextualize the PCC of 0.42 previously described as "moderate." We have removed this imprecise wording, as the PCC represents a correlation of dLFC values from different biological contexts (HCT 116-Cas9 and HCT 116-Cas12a cells) and falls within a normal range, as highlighted in the referenced work.

R2 response: The authors have addressed my concern.

2. AB-BA agreement. The authors report the number of possible pairs as $\{X \text{ choose } 2\}$. For instance, they report 210 unique pairs among 21 genes. However, I suspect that all gene pairs are targeted in two orientations of gRNAs on the plasmid: geneA -geneB (AB) and geneB-geneA (BA). If this indeed was the case, we had for the 210 pairs another 210 pairs where the orientation is inversed as well as 21 'self-interaction' scores (geneA-geneA, geneB-geneB, ...). While the latter can be an informative data quality metric too, I would especially emphasize that reporting the AB-BA agreement of genetic interaction scores is a crucial reproducibility metric. This will likely be different from the effect the orientation has on single KO effects reported in Figure EV2D (also since single KO effects have a different density and effect size).

Authors: The reviewer raises an important point about the AB-BA orientation in our screens. Indeed, the SLC×SLC screens are symmetric, with both orientations of gene pairs (AB and BA) present on the plasmid. However, the SLC×Enzyme screens are directionally paired, with SLCs as gene1 and Enzymes as gene2, making the AB-BA analysis applicable only to the SLC×SLC screens.

To address this point, we have added an AB-BA agreement analysis for both the Cas12a-SLCxSLC and Cas9-SLCxSLC screens as Fig. EV2E. In this analysis, we plotted the observed log fold changes (LFC_obs) of gene2-gene1 against gene1-gene2, using gene-level averages of the corresponding gRNA orientations. We chose to plot LFC_obs rather than genetic interaction scores (dLFC) because interactions are sparse, and plotting dLFC values would result in a difficult to interpret distribution of points around 0. Linear regression analysis on LFC_obs values shows good agreement between orientations, with the Cas12a system showing a slope of 0.95 and R2 of 0.82, and the Cas9 system showing a slope of 0.90 and R2 of 0.83 (Fig. EV2E). These results indicate that while the orientations agree well overall, double knockouts exhibit a slightly stronger guide position effect compared to single knockouts, despite the single knockout values being derived from guide pairings with cutting guides targeting olfactory receptors.

R2 response: The authors have partially addressed my concern. The additional illustration is helpful and the reasoning for why AB-BA comparisons of GIs are limited is not unexpected. At this point it is not entirely clear to me why there is low AB-BA correlation of GIs, but I expect a high false negative rate, which is rather usual for this type of data. Perhaps the authors could briefly mention false negatives as a possible source of limited AB-BA correlation.

3. Context similarity and differences. I would like to see a more extensive statistical assessment of genetic interactions common and private to the experimental conditions. More specifically, it would be important to learn how genetic interactions change in different conditions given the reproducibility in the same condition.

Authors: We thank the reviewer for an excellent question, as it touches on a central biological question: how genetic interactions respond to changes in the environment. While direct comparison between Cas9-SLCxSLC and Cas12a-SLCxSLC screens is limited by potential Cas system-specific effects, we can analyze interaction changes across glucose, hypoxia, galactose, and antimycin conditions in the Cas12a-SLCxEnzyme screen.

Of 134 single KO effects, 41 (31%) showed substantial changes (LFC > 1) in antimycin/glucose and/or galactose/glucose comparisons (Figure 3C). This high proportion likely reflects the enrichment of mitochondrial genes in our SLCxEnzyme library and the metabolic perturbations induced by galactose and antimycin, as we now note on page 8. Among 2,357 tested gene pairs, 186 (8%) showed significant interaction changes between conditions, with 0 changes in hypoxia, 47 in galactose, and 157 in antimycin.

This analysis combined week 3 and week 5 replicates (6 dLFC values per gene pair) for each condition, using the two-stage linear step-up procedure of Benjamini, Krieger, and Yekutieli for P value adjustment (thresholds: $P_{adj} < 0.1$, $|dLFC| > 0.3$). We visualize all condition-specific interaction changes in Figure Appendix S1, now referenced on page 9.

R2 response: The authors have addressed my concern.

Minor comments

1. It is unclear what the mentioned Pearson correlation coefficients 'across time points' (0.76-0.91) on page 5 exactly refer to. Figure EV1D is cited in this context but only shows one data population and the comparison does not compare time points. In this regard, time points of the same replicate in a CRISPR screen often tend to be very highly correlation.

Authors: We thank the reviewer for highlighting this ambiguity. The intent of this sentence was to highlight that the dLFC values from the smaller Cas12a-Metal-SLCxSLC screen were well recovered in the larger Cas12a-SLCxSLC screen. The Pearson correlation coefficients refer to comparisons between dLFC values from corresponding time points in the Cas12a-Metal-SLCxSLC and larger Cas12a-SLCxSLC screens (e.g., input vs. input, week 1 vs. week 1, ...). Note that the final time point was changed from week 4 in the Metal screen to week 3 in the larger screen. Thus, in Fig. EV1D, we compare Cas12a-Metal-SLCxSLC (week 4) with Cas12a-SLCxSLC (week 3). We have revised the text and figure caption to remove ambiguity and only refer to the specific correlation that is shown in Fig EV1D.

R2 response: The authors have addressed my concern.

2. Please add description of error bars in Figure 1E, F and Figure EV1A, B and all alike figures in the legends.

Authors: All error bars represent SD. This information was added to the relevant figure captions.

R2 response: The authors have addressed my concern.

3. Please label samples (week_x_x), units and the data represented in Figure EV1C. The legends mention gRNA abundance - is this normalized?

Authors: We added the labels to Figure EV1C. The figure caption now clarifies that the analysis was performed on raw read counts.

R2 response: The authors have addressed my concern.

4. Figure EV2B. Please move dot labels so they are readable.

Authors: We moved the dot labels to ensure readability.

R2 response: The authors have addressed my concern.

5. I would appreciate illustrations (e.g. in extended view figures) that show how single KO effects of each tested gene, such as the values on the x-axis of Figure EV2A, are scattered against the combinatorial effects of those genes with different query genes (I understand that the experimental gene matrix is symmetric and all genes can be considered a query gene too).

Authors: We thank the reviewer for this suggestion. While plotting single knockout effects directly against their combinatorial effects is feasible, this would show each interaction (AB) twice - once for gene A and once for gene B. To avoid this redundancy and potential misinterpretation from plotting a single experimental value (AB) twice, we have instead added Figure EV2F, which directly compares observed double knockout effects (AB) at week 3 (LFC_obs (week 3)) with expected effects calculated as sum of single knockout effects (A+B) (LFC_exp (week 3)). As such, each gene pair is plotted once, showing how combinatorial perturbations deviate from additive expectations based on single knockout measurements, and where interactions are scored in relation to single knockout magnitude.

R2 response: The authors have addressed my concern.

6. Please expand the description in the methods of how genetic interactions were scored including how many data points go into which statistical test.

Authors: We have revised the data analysis section (page 36) to provide a more detailed description of how genetic interactions were scored. Additionally, Table EV8 has been updated to clarify the number of data points used in statistical testing. For example, in the Cas12a-SLCxEnzyme screen, SLC-A_guide_1 was paired with Enzyme-B_guide_1, and SLC-A_guide_2 was paired with Enzyme-B_guide_2, with 6 sgRNAs per gene resulting in six guide combinations per gene pair. Across three replicates, this yielded 18 read counts for gene pairs, from which 18 log₂ fold changes (LFC_obs) values were calculated. These 18 LFC_obs were tested for significance against 18 LFC_exp values derived from the pairings of SLC-A and Enz-B sgRNAs with olfactory sgRNAs (i.e. single KO effects). These numbers are now explicitly listed for each screen in Table EV8.

R2 response: The authors have addressed my concern.

7. Please provide the code used for generating the genetic interaction scores.

Authors: We did not use code in the traditional sense for generating genetic interaction scores. Read processing was performed through the Galaxy platform (<https://usegalaxy.org/>) using algorithms as described in the methods section. Raw read counts were subsequently processed in Microsoft Excel to calculate the log₂ fold changes for each sgRNA pair as well as the mean single KO effects (calculated for each sgRNA pair using the corresponding control sgRNA pairs). We then performed the significance testing as described in the Method section and in Table EV8 using the student's t-test function in Microsoft Excel and performed p value adjustment with GraphPad Prism. We expanded the methods section to clarify how the interaction cores were generated.

R2 response: I appreciate the clarification. While I think that the availability of code that executes the data processing and assures reproducibility is crucial for data sets such as those presented here, I am convinced that this work will be of great value for the community as is. In absence of the requested code, I would strongly advise to make the data at all major intermediate processing steps available.

Changes to Manuscript (Editorial Requests 1-13):

1. Figures removed from manuscript (#1)
2. Author contributions removed (#3)
3. Appendix figure callouts corrected on p. 9 and p. 20 (#4)
4. Callout for Fig. EV3B corrected on p. 8 (#4)
5. Tables and datasets renumbered; titles and callouts updated (#5)
6. Legends added to all datasets (EV1-7) as the first sheet, and to Table EV1 in first row on same sheet (#5)
7. Competing interests heading renamed (#6)
8. Supplementary information heading removed (#7)
9. Data availability "Note" removed (#8)
10. Appendix PDF now provided in the correct PDF format (#9)
11. Synopsis resized (#10)
12. Methods heading corrected (#11)
13. Legends moved after references (#1, 12)
14. Headings renamed and reordered (#12)
15. Appendix PDF legend removed from "Table(s)" headings
16. Added missing information on significance tests, n-values, center of error bars, and scale bars (#13)

Additional Corrections:

1. Table EV1: Corrected the mistake in cell D11 from "per replicate of w2,3,5" to "per replicate of w1,4,5"
2. Removed duplicated "Error bars SD" from the EV Fig 2D legend

Response to partially addressed peer reviewer comments:

1. In response to the only partially addressed concern raised by Reviewer 2 (Major Comment #2), we have added the following sentence on p. 6: *"These double KO correlation values are lower than those previously reported for paralog pairs using Cas12a ($R^2 = 0.86$) and Cas9 ($R^2 = 0.96$) (Li et al., 2022), potentially due to false negatives in our screen."*
2. In response the only partially addressed concern raised by Reviewer 2 (Major Comment #7),, we have updated our GEO submission GSE269905 (<https://www.ncbi.nlm.nih.gov/geo/query/acc.cgi?acc=GSE269905>). Given the large file size of

the tables, we decided to upload them to GEO rather than uploading them to MSB. The GEO submission now contains 5 additional tables (one per screen) representing a major intermediate processing step. These tables include all guide combinations present in each screen, along with calculated log2 fold changes (LFC_obs, LFC_gRNA1, LFC_gRNA2, LFC_exp, and LFC_obs-exp). As such, the raw data and the intermediate processing step is now deposited at GEO, and the final processed data at gene-level is provided as Dataset EVs along with the paper.

Added tables to GSE269905:

GSE269905_LFC_per_sgRNA_pair_Cas12a-Metal_SLCxSLC.xlsx

GSE269905_LFC_per_sgRNA_pair_Cas12a-SLCxEnzyme.xlsx

GSE269905_LFC_per_sgRNA_pair_Cas12a-SLCxSLC.xlsx

GSE269905_LFC_per_sgRNA_pair_Cas9-SLCxEnzyme.xlsx

GSE269905_LFC_per_sgRNA_pair_Cas9-SLCxSLC.xlsx

31st Mar 2025

Manuscript number: MSB-2024-12685RR

Title: The genetic interaction map of the human solute carrier superfamily

Dear Giulio,

Thank you again for sending us your revised manuscript. We are now satisfied with the modifications made and I am pleased to inform you that your paper has been accepted for publication.

Yours sincerely,
Jingyi

Jingyi Hou, PhD
Senior Editor
Molecular Systems Biology
